# Current and Future Prospective of Injectable Hydrogels—Design Challenges and Limitations

**DOI:** 10.3390/ph15030371

**Published:** 2022-03-18

**Authors:** Saud Almawash, Shaaban K. Osman, Gulam Mustafa, Mohamed A. El Hamd

**Affiliations:** 1Department of Pharmaceutical Sciences, College of Pharmacy, Shaqra University, Shaqra 11961, Saudi Arabia; gulam@su.edu.sa (G.M.); aboelhamdmohamed@su.edu.sa (M.A.E.H.); 2Department of Pharmaceutics and Pharmaceutical Technology, Faculty of Pharmacy, Al-Azhar University, Assiut 71524, Egypt; shaabanosman@azhar.edu.eg; 3Department of Pharmaceutical Analytical Chemistry, Faculty of Pharmacy, South Valley University, Qena 83523, Egypt

**Keywords:** injectable hydrogels, biodegradable polymers, chemical and physical crosslinking, tissue engineering

## Abstract

Injectable hydrogels (IHs) are smart biomaterials and are the most widely investigated and versatile technologies, which can be either implanted or inserted into living bodies with minimal invasion. Their unique features, tunable structure and stimuli-responsive biodegradation properties make these IHs promising in many biomedical applications, including tissue engineering, regenerative medicines, implants, drug/protein/gene delivery, cancer treatment, aesthetic corrections and spinal fusions. In this review, we comprehensively analyze the current development of several important types of IHs, including all those that have received FDA approval, are under clinical trials or are available commercially on the market. We also analyze the structural chemistry, synthesis, bonding, chemical/physical crosslinking and responsive release in association with current prospective research. Finally, we also review IHs’ associated future prospects, hurdles, limitations and challenges in their development, fabrication, synthesis, in situ applications and regulatory affairs.

## 1. Introduction

Biomaterial with desirable therapeutic efficacy, drug-delivery capabilities and enhanced properties has paved the way for the effective targeted treatment of diseases and injured tissue repair. It has become a promising area of human health-related research in biomedical material, agricultural, environmental, chemical, engineering and other applied sciences [1]. Over the last decade, hydrogels and their desirable biomedical applications, such as biocompatibility, non-toxicity, biodegradability, flexibility, biofunctionality, sol–gel transitions, tunable properties, physico-chemical properties, degree of swelling and porosity, synthesis, crosslinking, drug-encapsulation, sustained/control release of therapeutic agents (drug, proteins, genes, cells, etc.), tissue regeneration/repairing, wound healing, cancer chemotherapy and the treatment of superbug bacterial infections, have been extensively observed and investigated [1,2,3,4]. Hydrogels are a hydrophilic polymer system that has the capability of retaining a significant amount of water and swelling in an aqueous medium. However, due to their three-dimensional (3D) crosslinking structure, they remain insoluble in water and biological fluids [1,4]. The high water content and their polymeric meshwork and side chain maintain their well-defined structures due to their highly adsorptive surfaces, as well as spatiotemporal control over their physico-chemical properties and the release of encapsulated drug from hydrogel cores [5].

The typical administration of a hydrogel formulation begins from its early use in contact lenses to its extensively developed complicated applications, mostly in tissue engineering and gene/DNA delivery for sustained and controlled drug delivery. Wound healing dressing, sensors, mucoadhesive, bioactive factor delivery, bug targeting and cancer chemotherapy are other typical applications [6]. Extensive research on hydrogel formulations and dosage-form preferences has widened their biomedical applications and availability to consumers. IHs are gaining traction over the traditional intravenous therapies, which have many systemic toxicities in comparison, including myelosuppression, liver or kidney dysfunctions, non-targeted delivery, prolonged/controlled release and neuro- and other organ-toxicities. IHs can competently avoid these problems by releasing drugs at the infective/tumor site or target sites with localized drug toxicities [2,4,7].

IHs possesses a unique biocompatibility and hydrophilicity, as well as a phase transitions ability—sol (liquid) to gel (solid) to form solid-like gel states to administer and assist in the encapsulation and release of drugs, genes, DNA, proteins and cells in a sustained and controlled manner [8]. They are prepared from natural and synthetic polymers through various mechanisms, such as physical and chemical crosslinking methods (discussed in 1.2 classification and synthesis) into different types with their respective advantages and disadvantages [7]. Some of the characteristics associated with IHs are shown in Figure 1 [9], whereas Figure 2 [10] shows the designated and fabricated IHs for photothermal antitumor therapy.

### 1.1. History of Biodegradable Hydrogel Formulations

In 1960, Whichterler and Lim synthesized and reported the first hydrogel poly (2-hydroxethyl methacrylate) (PHEMA) and employed its uses in contact lenses, which have the ability of absorbing moisture. Its 3D crosslinking structure demonstrates and resembles modern hydrogels. Its discovery fascinated and attracted the interest of the scientific research community [8]. Then, in the 1980s, Lim and Sun fabricated calcium–alginate gel composites for cell embedding with islet-droplet microcapsules and confirmed that these hydrogels had similar physical characteristics to biological soft tissues. Later on, hydrogel became a hotspot research subject for biomaterial scientists, where then numerous natural, synthetic and hybrid polymer hydrogels were extensively synthesized and studied [4]. The sol–gel phase transition properties, mucoadhesiveness and biodegradation through physiological stimuli (pH, enzyme and temperature) of these polymeric materials revolutionized the whole field because they were non-invasive therapeutic agents that provided a controlled/sustained delivery depot at the site of interest [11,12,13].

### 1.2. Why Hydrogels?

As we know, a hydrogel has a 3D structure containing a high amount of water or biological fluid, which makes an absorbable and firm hydrophilic system. Its polymer chemical nature (formed through physical or chemical crosslinking), structural morphology and hydro-porous state of equilibrium provide advantageous properties, such as mechanical robustness, flexibility, external/internal transport, biocompatibility, biodegradability and the ability to simulate the extracellular matrix environment. These promising distinctions make hydrogels a potential candidate for many biomedical and therapeutic applications [4]. The water-absorbable tendency phenomenon in hydrogels, swelling media and crosslinked bonding strength are all due to the presence of polar hydrophilic functional groups, such as OH, SO3H, COOH, NH2 and CONH2. These hydrophilic groups in hydrogel polymers play a major role in their secondary interactions with physiological tissues and participation in water uptake. Furthermore, IHs’ in situ controlled gelation transformations, process procedure kinetics, injectability, 3D carrier properties, biocompatibility, non-invasiveness and shape-adaptation make them distinctive in biomedicines, cancer chemotherapies, tissue engineering and drug delivery. However, along with these applications and advantages, they also have to face some challenges and limitations to meet various clinical requirements [14]. Figure 3 [11,15] depicts some of the characteristics associated with IHs’ physico-chemical properties, sol–gel transitions, gel formation underneath the skin in an animal model and stem cell therapy applications. 

The fabrication of an injectable hydrogel from the lab to industrial process needs quality design approaches with critical quality attributes, as mentioned by [9]. Figure 4 summarizes all these parameters connected with the features of the starting material (active ingredients, biopolymers, solvents and components ratios) throughout the stages of the fabrication process (i.e., heating temperature, agitation speeds and reaction time), which are outlined in the Ishikawa model diagram to potentially modify these formulation steps.

### 1.3. Classification of Hydrogels

For convenience, hydrogels can be classified into different categories, as mentioned in Table 1. On the basis of the nature of the material, gelation mechanism, biodegradability, side group characteristics and attachment, degree of porosity and swelling, they were basically classified into natural/synthetic/hybrid hydrogels, structural morphology, crosslinked hydrogels, charge (anionic, cationic or neutral) hydrogels, biodegradable/non-biodegradable, low/high swelling or superabsorbent hydrogels, micro/macro or super-porous hydrogels, etc., [1,4]. These various hydrogel systems have wide applications in biomedical sciences through various means. They each have advantages and disadvantages according to their fabrication and structural design—for instance, uncontrolled biodegradation and batch-to-batch variations associated with natural polymeric material make it difficult to control their mechanical strength and properties, which in comparison with synthetic polymers usually have a well-defined 3D structure and robustness [16,17,18,19,20]. 

### 1.4. Synthesis and Generation of Hydrogels

Different techniques have been utilized to generate 3D hydrogel structures either from natural or synthetic sources via chemical crosslinking (click chemistry, photo polymerization, Schiff’s base, enzyme-catalyzed or thiol-based Michael reactions) or physical crosslinking (induced through temperature, pH, ionic interactions, stereo complexation or guest–host inclusion reactions) [17,18,19,20]. Hydrogels generated through these methods have the capabilities of the encapsulation and release of drugs and biomolecules (DNA, gene, RNA and proteins) for various therapeutic purposes [21]. IHs that are generated through physical crosslinking generally have poor mechanical properties in comparison with chemical crosslinking hydrogels. It should be noted that chemically crosslinked hydrogel synthesis has slow gelation kinetics, so on injection it generates hydrogels in situ immediately. Recently, dual-gelling hydrogels have been developed, which have the combined mechanical strength capabilities of a chemical gel and fast gel formability of a physical gel [8,11]. Figure 5 shows various examples of crosslinking polymers (chemical and physical) [22].

#### 1.4.1. Chemical Crosslinking

The hydrogel class that can undergo the transition phase, in which it changes from a liquid state to a gel state (sol–gel transition) by forming new covalent bonds in a polymer meshwork through certain chemical reactions, is called chemically crosslinked hydrogels (Figure 5). They have extensive applications in implants and injectable devices, where in situ gels are desired [4,19]. Chemically crosslinked reactions are shown in Figure 6.

##### Photo-Crosslinked Polymerization

Photo-crosslinked hydrogels can be generated from polymers (in an aqueous solution) that contain photosensitive molecules and polymerization catalysts. When exposed to UV/visible light (an external stimuli) systems, the photo-sensitive molecule decomposes, polymerizes and generates free-radicals—for example, methacrylate (acrylate groups), which upon irradiation undergoes rapid polymerization (Figure 6A) [8]. Moreover, photo-polymerization has several advantages, such as controlled gelation kinetics, patterned 3D structured hydrogels for release studies, low energy, no excessive heating or local toxicity, free solvent requirement and rapid reactions to mild conditions [6]. An initial example of an injectable photo-crosslinked hydrogel for a biomolecule delivery system was introduced by Hubbell et al. (1993) [23,24].

##### Click Chemistry

The high specificity, biorthogonality, excellent efficiency and short reaction to mild conditions of click chemistry make it of great interest as an emerging platform for macromolecule delivery and tissue engineering. It is considered to have a significant role in the fabrication of hydrogels, nanogels and microgels, as well as in bioconjugation and polymer synthesis. Examples of click chemistry reactions include Cu (I)-catalyzed alkyne-azide cycloaddition reactions (Figure 6B), Diels–Alder cycloaddition radical-mediated thiol-Michael reactions, catalyst-free alkyne-azide coupling reactions and Schiff’s base reactions. High efficiency under various physiological conditions and extreme chemo-selectivity are its key advantages. 

In click chemistry reactions, the azide and alkyne groups were introduced into polyvinyl alcohol (PVA) and polyethylene glycol (PEG) chains via its carbamate linkage to generate a hydrogel 3D meshwork. This took place by the mixing of alkyne-PVA either with azide-PVA or with PEG diazide, which can lead to the generation of hydrogels in the presence of a Cu (I) catalyst. The mixing of these two multi-functionalized PVAs (alkyne-PVA and azide-PVA) has proven to be more effective in gel formation than using a bi-functional crosslinker. Gelation ability and crosslinking density were enhanced with an increasing concentration of the functional groups.

##### Schiff’s Base Reaction

Schiff’s base chemical crosslink compounds (usually containing a double bond of carbon and nitrogen) can be obtained when nucleophilic amines or hydrazides react with electrophilic carbon atoms of aldehydes or ketones. Schiff’s base reactions can occur without any chemical or catalyst under certain biological conditions (Figure 6C). These characteristics have attracted significant concerns for the fabrication and generation of in situ formed IHs, which will have a control reaction rate within the respective pH medium.

##### Enzyme-Catalyzed Reactions

Enzymatic-catalyzed crosslinking reactions occur biologically in the presence of enzymes. Mild reaction conditions are required, including a neutral pH, temperature, substrate specificity and aqueous environment, which can prevent toxicity. Transglutaminase, horseradish peroxidase, phosphopantetheinyl transferase, tyrosinase and lysyl oxidases have been used to prepare hydrogels through enzyme catalysis, particularly for tissue engineering. In enzyme-catalyzed crosslinked hydrogels, horseradish peroxidase (HPR) has been the most striking candidate due to its fast gelation, mechanical strength, high stability and easy purification [8,25]. Figure 6D depicts an HPR-catalyzed oxidative coupling of phenol components in the presence of hydrogen peroxide (H2O2), which acts as an oxidant. These oxidative catalysts have been extensively applied in the generation of hydrogel systems from natural polymers, such as gelatin, dextran, hyaluronic acid and chitosan [8], which are predominantly applied in tissue engineering/repairing [8,26] and protein delivery [27]. 

##### Thiol-Based Michael Reaction

The nucleophilic addition of a nucleophile to an α- or β-unsaturated carbonyl compound is referred to as a Michael reaction, while its chemical crosslinking when the nucleophile components are thiol- and amine-bearing molecules, where unsaturated carbonyl components are commonly attached to acrylate/methacrylate and vinyl sulfone groups, is known as a thiol-based Michael reaction. They have been extensively applied in biomedicine, optoelectronics, pharmaceutics, composites, coatings and adhesives due to their controlled reaction rate, optimized conditions, relative inertness with biomolecules and high percentage chemical yields. These thiol-based Michael reactions are used to generate in situ crosslinked IHs for biomedical applications in protein delivery and tissue engineering (Figure 7) [28]. When a dithiol-PEG solution is mixed with a dithiol-PEG at an equal ratio, gelation can be obtained within 34–40 s, though this depends on the PEG solution, functionality and polymer concentration [8]. 

#### 1.4.2. Physical Crosslinking

Physically crosslinked hydrogel achieves a gel state by changing its intermolecular forces, such as hydrogen bonding, hydrophobic interactions and electrostatic ionic forces. It can also be attained through intermolecular assemblies’ guest–host interactions, complementary binding and stereo-complexation, etc., [29]. These changes are driven either by the internal arrangement of the polymers themselves or may be due to some external stimuli, such as ionic strength, heat, light, pH, pressure, sound and electric field or due to the presence of certain molecules (Figure 8) [30]. Physically crosslinked hydrogel possesses a specific gelation time, mechanical strength and biodegradation mechanism. Figure 9 represents physically crosslinked reactions and their molecular-level gelation mechanism and concept [8].

##### Temperature-Induced

Temperature induction can cause changes in the solubility of the polymer and its 3D structure, which may cause a sol–gel phase transition (Figure 3A,B). Fast gelation in hydrogel synthesis can be achieved through temperature changes. The temperature-sensitive hydrophilic–hydrophobic hydrogels, in response to changes in temperature in an aqueous solution, determine the macroscopic soluble–insoluble transition state, as shown in Figure 9A. On the basis of temperatures changes, these hydrogels can be further classified into different temperature-responsive and -sensitive groups.

##### pH-Induced

In most specific applications of thermal-sensitive hydrogels, possible blockade during an injection, a lack of ionic interaction in the protein/drug/gene delivery, striving in dissolving and material storage limit its utilization in hydrogel preparations. The limitations of these hydrogels led to the development of pH-sensitive or combined pH/temperature-sensitive hydrogels. As we know, each system in the human body possesses a different physiological pH environment, such as the stomach (pH 1.35–3.5), liver (pH 7.6–8.8), intestine (pH 6–7.5), tumor sites (pH 6.4–7.0), blood vessel (pH 7.35–7.45) and vagina (pH 3.8–4.5), which can be useful for stimuli-sensitive responsive systems for the delivery of bioactive agents. The pH-responsive phase transition between soluble and insoluble mainly occurs via protonation/deprotonation through an ionization constant (pKa) among ionized groups (Figure 9B). It has been observed that pH-sensitive or responsive polymers are weak polyelectrolytes, which is supposed to be due to their acidic moieties such as sulfonamides, carboxylic acid or basic tertiary amine groups that ionize either at a low or high pH.

##### Ionic Interactions

In the ion-gelation mechanism, pH-responsive hydrogel can be generated via ionization and deionization, whereas ion-induced complex hydrogels can be synthesized through the opposite electrostatic ionic interactions (Figure 9C). 

##### Guest–Host Inclusion

When chains or particular groups from guest molecules are placed into the cavity of cyclodextrin (CD), which acts as a host molecule to form a supramolecular inclusion complex, this is termed guest–host inclusion physical crosslinking (Figure 9D). Cyclodextrin and its host supramolecular complex geometry have led to an expansive application range in ophthalmic/nasal drugs and peptide/protein delivery. 

##### Stereo-Complexation

Polymer of opposite chirality can result in a physically crosslinked hydrogel through a stereo-complexation gelation mechanism (Figure 10). For instance, when an enantiomer D-lactic acid and L-lactic are mixed together, a PLA stereo complex is formed, which is a selective stereo interlocking system with significant mechanical strength, melting point and hydrolytic stability in comparison with its precursors. The stereo-complexation has been found in various types of polymer pairs, but its biodegradable (poly-methyl methacrylate) and non-biodegradable (poly-lactic acid) interlocking is the most prevalent utilization.

##### Complementary Binding

Peptide-based hydrogels can be prepared through the association of complementary β-sheet subjects, β-hairpin or the coiled-coiled assembling of α-helix subjects. This mechanism can also be used for self-assembled DNA hydrogels, prepared via base-pairing complementary DNA strand interactions. Similarly, the generation of biomolecule-responsive bioconjugated hydrogels was developed through a ligand–receptor interaction between growth-factor/heparin, concanavalin A/glucose or antigen and antibody binding between rabbit IgG and goat anti-rabbit IgG. Finally, for the production of metallo-hydrogels, metal–ligand coordination was projected either among iron (II)/bipyridine, iron (III)/catechol oppressions or nickel (II)/terpyridine [8].

### 1.5. Surface Chemistry, Internal Bonding and Characterization

The internal bonding strength and surface chemistry of hydrogels depend on polymer crosslinking (chemical/physical). Most chemically crosslinked hydrogels have covalent bonds, resulting in strong irreversible gels due to their mechanical bonding strength and potential harmful properties that make them unfavorable for biomedical applications. In contrast, physically crosslinked hydrogels have non-covalent bonds (most have ionic interactions, hydrophobic interactions and chain entanglement) show promising biocompatibility, are non-toxic, have reversible responsiveness by changing the external stimulus (light, heat, pH, temperature, etc.) and thus can be extended to biomedical applications [29,31].

During the injection, at target sites, the hydrogels undergo a rapid sol–gel phase transition, which allows the matrix to easily take the shape of the cavity, providing a good fit and interface in tissues. Moreover, different therapeutic molecules and even cells can be assimilated by simple mixing with gel solution prior to injection [32]. The covalent bond in chemical crosslinking forms stronger ionic bonds with various functional groups of crosslinking agents introduced. Thus, during the polymerization process (chemical/physical), different polymeric structures, such as homo-polymers and linear-, block- or graft- copolymers, are formed. These hydrogels are used in solid-molded form (contact lenses), as microparticles (as a bioadhesive or wound dressing and treatment), as constrained powder matrices (pills or capsules), as coatings (for catheters and implant devices), as membranes or sheets (as a reservoir in a transdermal drug patches), as solid encapsulations (in osmotic-pumps) and as liquids, which undergo gelation upon heating or cooling [33]. Various crosslinking strategies and internal bonding characterizations are shown in Figure 11 and Figure 12 [34,35]. 

### 1.6. Commercially Available Hydrogel-Based Dosage Forms

Oral hydrogel formulations are generally prepared as a reservoir system or matrix, in which the dispersed drug diffuses out after it is subjected to aqueous medium upon subsequent dissolution. Here, in this system, the variation in drug/macromolecule delivery kinetic rates is associated with physico-chemical properties, dosage form variability and polymers (including thickness). 

Similarly, ophthalmic hydrogels and their current delivery/carrier systems are also challenging formulations in future prospects due to the unique anatomical, physiological and biochemical barriers of the eyes. Ocular hydrogels should be biocompatible, non-toxic, biodegradable with prolonged retention time, non-invasive and easy to prepare. 

Hydrogels used for wound dressing possess distinctive advantages of absorptive surfaces and retaining a desired moisture level within the wound region. It also permits proper moisture and oxygen exchange in the wound surroundings; biocompatibility; the retention of tissue structure adaptation; ease of application due to softness, elasticity and flexibility; soothing sensations; capability of absorbing serous discharge/pus from wounds; and decreased wound healing process interference. All these associated features make hydrogels ideal devices for wound dressing and healing, which are well tolerated by patients and accepted by physicians. Table 2 shows some of the commercially available hydrogel dosage forms on the market for oral and ocular delivery and for wound dressing [31].

### 1.7. Responsive Released Studies

The drug release from the hydrogel systems mainly occurs through the shrinking or swelling of the hydrogels and then their diffusion through the polymer network, which determines the rate of swelling and controlled release. When these kinds of hydrogel properties, such as swelling behavior, structure elucidations, mechanical strength or permeability, can change in response to different stimuli, they are termed “responsive stimuli” hydrogels or environmentally sensitive hydrogels. These stimuli can be used effectively for various controlled-hydrogel delivery systems (Figure 13) [36]. Stimuli-responsive polymers (also known as smart polymers) can be used for various kinds of responsive hydrogel systems. They can be prepared either using these smart polymers or by modifying polymer structures with smart polymers and making them responsive to certain environmental stimuli/triggers. These external environmental stimuli may be the pH, temperature, ionic concentration or volume transitions (collapse or phase transitions) of the hydrogels, which are associated with and depend on their swelling properties. The internal bonding and crosslinks make the hydrogel water insoluble, which allows proper geometrical dimensions to be attained. Thus, these changes in the 3D structure and volume of hydrogels make them an attractive material for therapeutic delivery and tissue regeneration due to their low toxicity, good injectability, biodegradability, mucoadhesive and bioadhesive properties. As hydrogels are inert and show negligible inflammatory responses, thrombosis and tissue damage, they can be inserted into a precise shape for extended and prolonged use [36,37,38]. Examples of polymers that are responsive to stimuli are poly-N-isopropylacrylamide and pluronics (temperature sensitive), alginate (enzyme and ionic concentration), chitosan (pH and enzyme), etc. Some of these external/environmental stimuli mechanism responses are discussed in brief. 

#### 1.7.1. Thermosensitive

Thermosensitive hydrogels undergo a sol-to-gel phase transition in response to temperature variations. All their basic concepts, mechanisms and properties have been introduced and summarized recently in several review articles [38,39,40,41,42]. Novel thermosensitive hydrogel copolymers composed of poly (ethylene oxide) (PEO) and poly(L-lactic acid) (PLLA) were synthesized by Dr. Sung Wan Kim and coworkers in 1997 [9]. Other thermosensitive hydrogels that were reported from their work, such as poly(*N*-isopropyl acrylamide) (PNIPAM) and PEO-b-poly (propylene oxide)-b-PEO (PEO-b-PPO-b-PEO), are all PLLA-based thermosensitive IHs, which show biodegradability and biocompatibility with extended biomedical applications. Their work on thermosensitive responses has gained tremendous attraction in material sciences. In their follow-up research, Dr. Kim’s group worked on polymers and their gel formation upon decreasing temperature, even from an elevated temperature to body temperature or from room temperature to body temperature—for instance, PEO/poly(lactic-co-glycolic acid) (PLGA) and PEG/PCL, which can be employed as depots for controlled insulin release [7]. These thermosensitive responses can effectively protect heat-sensitive drugs from degradation through their spatiotemporal control over the drug release mechanism with a lower critical solution temperature (LCST) [5]. Thus, they have the unique property of being a free-flowing liquid at room temperature, turning into a viscous gel at body temperature, which makes them non-invasive and able to be administered in their liquid state anywhere within the body using a small needle without requiring major surgery [43].

#### 1.7.2. Temperature Sensitive

When thermoresponsive polymers, such as pluronics, are combined with natural polysaccharides, it can form a temperature-sensitive hydrogel, which can undergo a sol–gel phase transition between room temperature and ambient body temperature (Figure 14). The therapeutically released kinetics from these stimulus-responsive smart hydrogels can be controlled either by an ambient condition or external stimuli at the administration site, such as redox potential, low pH, disease state or the presence of certain enzymes. These stimulus-responsive hydrogel systems are either single responsive, dual responsive or multi-responsive during therapeutic delivery and tissue regeneration applications. 

Many polymers/hydrogels (chitosan, hyaluronic acid, agarose, etc.) exhibit a temperature-dependent sol–gel phase transition. These critical temperatures can be classified as either an upper critical solution temperature (UCST) or lower critical solution temperature (LCST). At UCST, the water solubility of polymers increases with higher temperatures, which increases their swell ratio and can lead to UCST hydrogel transitions from a gel (solid) to a solution (liquid) while they are below their UCST. However, polymers with LCSTs display an inverse or a negative temperature dependence, decreased solubility and a decreased swell ratio, while their transition from a liquid solution to a gel occurs as the temperature increases. For instance, the IH poly(ε-caprolactone-co-lactide)-b-poly-(ethylene glycol)-b-poly(ε-caprolactone-co-lactide) (PCLA-b-PEG-b-PCLA), which is an amphiphilic triblock copolymer, is a temperature-sensitive, tunable and widely utilized biodegradable polymer. It has wide applications in cancer immunotherapy, enhanced immune responses and increased immunogenicity in cancer vaccines [36].

#### 1.7.3. pH-Sensitive

Polymers in which changes in the hydrogel volume depend on the ionic strength and pH of the external environment are called pH-sensitive or responsive hydrogels, which contain protonated or readily hydrolyzable acids and bases such as amino and carboxylic groups. The extent of dissociation in these groups is responsive to external pH, causing the external or internal ion concentration to change, which also degrades the corresponding hydrogen in the gel. These conformational variations will reduce the crosslinking points in the gel structure and will show changes in the degree of swelling of the hydrogel, which can appropriately adjust and control the diffusion rate and drug release [36,38,41,42,44]. For example, let us consider chitosan, which is considered a pH-sensitive polymer and has the capability to show dissolution at a lower pH < 6.2 (mild acids) via amino group protonation, while at a higher pH, its cationic nature favors gel formation upon the neutralization of the repulsive electrostatic forces, or electrostatic interaction or interaction with hydrophobic moieties [36].

#### 1.7.4. Photosensitive

Light-sensitive hydrogels can be generated by incorporating light-sensitive moieties or chromophores, such as spiropyran, azobenzenes, o-naphthoquinone, anthracene, coumarin and nitrophenyl into their 3D polysaccharide structures [37,38,40,45]. According to photosensitive material characteristics, there are two kinds of mechanisms: first, the addition of photosensitive material to a temperature-responsive gel, where the light energy is converted into heat energy, which in turn makes the gel temperature reach the phase transition temperature; second, photosensitive materials are directly added to the gel structure, and upon exposure to light stimulation or under UV light/near infrared light, the ester groups degrade and photosensitivity converts the hydrophobic molecular endings into hydrophilic molecular endings, which in turn dissociate gel at the respective sites for drug delivery [36]. 

#### 1.7.5. Enzyme Sensitive

Similarly, enzymes can also be used for environmental stimuli or responsiveness for injectable hydrogel preparations. Most natural or synthetic polymers or their crosslinkings can easily be degraded through an enzyme hyaluronidase and metalloproteinase (MMPs) matrix, which cleaves the glycosidic linkages of polysaccharides/polymers. As we know, in some disease conditions, some of these enzymes are overexpressed at the site of infection, which helps in the dissociation of IHs—for instance, in the case of MMPs, which are function- and structure-related endopeptidases and are profuse in cancerous tissues [36]. 

#### 1.7.6. Dual-Sensitive Hydrogel

Among multi-sensitive hydrogels, dual-sensitive hydrogel systems with increasing demands for precise controlled release drugs have received more attention—for instance, co-sensitive hydrogels for pH and temperature. These co-sensitive hydrogels consist of a pH-responsive and temperature-sensitive hydrophilic polymeric meshwork [20,33]. A Schiff’s base and phenylboronate ester using phenylboronic-modified chitosan, poly(vinyl alcohol) and benzaldehyde-capped poly-(ethylene glycol) pH and glucose dual-responsive crosslinked IHs, which were prepared for protein drugs and live cell delivery for sustained drug/protein release, were used as a bioactive dressing for diabetic wound healing [46]. Similarly, another Schiff’s base crosslinking oxidized carboxymethyl cellulose with 3,3′-dithiobis (propionohydrazide) IHs was prepared for pH and redox dual stimuli responsiveness, which is characterized by tunable gelation, high swelling and low degradation kinetics. A bovine serum albumin (BSA) was used as a model drug, which shows a sustained release at pH = 7.4, and enhanced release at pH = 5.0 or in a reducing environment [47]. Another smart dual-responsive HIS of pH and reactive oxygen species (ROS) was prepared, which shows disease–microenvironment stimuli responsiveness. Grafting phenyl boronic acid to an alginate polymer side chain synthesized a highly specific dual-responsive hydrogel at low pH and high ROS. It showed drug release characteristics at the inflammation site and self-healing and remodeling capacity with antibacterial and anti-inflammatory properties, respectively, via the effective micelle preloaded encapsulation of the antibiotic amikacin and anti-inflammatory drug naproxen [21]. Similarly, in another study, dual-responsive thermal and pH-responsive hydrogels were fabricated and encapsulated with doxorubicin as a potential therapy for breast cancers [48].

#### 1.7.7. Glucose

As we know that skin serves as a body’s defensive line, it can easily be attacked and injured by infective agents. It also has a high healing capacity, and the generation of scar tissue may not be avoided, particularly when the injured area is large or deep into the skin (dermis/sub-dermal layers)—for example, wounds from a burn victim or diabetic patient, where chronic wounds and a high blood glucose level cause microvascular endothelial injury and vascular diastolic dysfunction. Thus, glucose-responsive IHs (carrying both insulin and fibroblast L929) are preferable, which are sensitive to a high glucose level. This system releases peptides faster at a high glucose concentration and allows the propagation of the cells in the gel matrix [3,45]. 

### 1.8. Sol–Gel Transition State of IHs

A phase transition in a polymer crosslinking, from liquid to solid at a critical point, is called the sol–gel transition state. It is believed that the sol–gel transition is carried out by two parameters, including the connectivity and polymer concentration. It is often understood in the context of a lattice-based percolation model. When the crosslinking of a polymeric liquid occurs, a sol–gel transition with an infinite 3D polymer network is produced at a critical point and in the presence of physical power law parameters, such as cluster growth, viscosity, elastic modulus and viscoelastic properties, in association with the bond and site percolation. In a bond-percolation model situation, all the lattice sites are occupied, and bonds between the neighboring sites are formed at random. It is often used to predict the gelation process from a molten and concentrated polymer system. However, its applicability to a dilute system is limited as a massive amount of diluent exists in the system where monomer units do not fill the space. Similarly, a site-percolation model situation involves sites that are partly occupied, and bonds between the occupied sites are formed. It seems to predict the sol–gel transitions of dilute systems [49,50]. 

The critical sol–gel transition patterns and concentration of PCLA (poly-caprolactone-co-lactic acid) and alginate–PCLA bioconjugates were found to be 17 wt.% and 16.8 wt.%, respectively. In aqueous solutions, these copolymers assembled into flower-like micelles with PCLA blocks and PEG blocks as cores and shells, respectively. These concentrated clusters or micelles induced gelation at body temperature (Figure 15), which is mainly due the hydrogen bonding between hydrophilic copolymers and water. However, under the physiological condition, the hydrophobicity increased due to the weakening of hydrogen bonds, resulting in micelle aggregation and the formation of rigid bridges between hydrophobic PCLA segments, which led to gel formation. When the temperature increased further, hydrophobic blocks shrunk, and the PEG blocks underwent dehydration, which led to phase separation [13]. 

## 2. Current Trends

### 2.1. IHs and Its Application in Drug Delivery System and Biomedical Engineering Applications

The current trends in biodegradable injectable hydrogel research and its wide application in the biomedical field make it one of the most important therapeutic formulations in drug delivery systems and regenerative medicines—for instance, its application in cartilage regenerations, where its physical properties can be designed such that they can properly match with articular cartilages, in association with their mechanical robustness and scaffold with native tissues. The current research diversifies, proceeds and focuses on the application of gene/drug/DNA/protein delivery, cancer therapy, immunotherapy, tissue regeneration (Figure 16) and engineering, vaccine delivery, implants, stem cell therapy, wound healing and many more—e.g., controlled, prolonged and sustained release formulations [9,12,13,14,20,25,29,34,43,51,52,53,54].

### 2.2. Drug Delivery

As we know, conventional drug delivery approaches are associated with undesirable drug absorption, rapid metabolism, repeated dosing, systemic toxicities and easy degradation under certain physiological conditions. Injectable hydrogel with sustained drug release properties, degradability and tunable physical properties make it a promising smart drug delivery system, which can overcome and optimize all those disadvantages. Biocompatibility, good syringeability with minimal invasiveness and responsiveness to environmental or enzymatic stimuli make these IHs smart therapeutic systems [41]. In comparison with other systemic or topical administrations, drugs using IHs are promising for controlled biodistribution and sustained release purposes. The 3D structure and water dispersed phase hydrogels can encapsulate water-soluble drugs, though water-insoluble drugs can also be loaded. Macromolecular compounds or bioactive molecules, such as genes, peptides, proteins or even living cells or organs can also be entrapped in hydrogels [14,54,55,56]. Figure 17 [57] and Figure 18 [58] illustrate drug delivery of IHs.

#### 2.2.1. Protein Delivery

Protein conjugation, assembly, genetic sequencing and diverse functionality have created a venue for diverse bioactive protein hydrogels—for instance, covalently bonded peptide/protein pairs, SpyTag/SpyCatcher or SnoopTag/SnoopCatcher, which are genetically encoded through click chemistry by forming isopeptide bonds, which are associated with the foundations of various crosslinker hydrogels and soft tissue material. These approaches are more advantageous than the hydrogel generation from natural or synthetic polymers in cyto-compatibilities, biofunctionalization, fabrication, biodegradability and biocompatibility. Thus, these stimuli-responsive polypeptides are now used in developing artificial protein polymer devices for drug/protein delivery, cancer chemotherapy and diagnostics. Similarly, metal–ligand coordination and its transcriptional regulation, under water adhesion and protein phase transition, are also gaining considerable attention. 

The chemical structure and molecular weights of polypeptide hydrogels can be adjusted through various crosslinking methods. These transformations (reversible, hydrophilic, hydrophobic, etc.) can be confirmed through endogenous responses, such as pH, enzymes and reduction, or through external stimuli, such as light, heat and temperature. These stimuli-responsive polypeptides have been extensively used for the fabrication of various drug carriers comprising nanogels, hydrogels and micelles for various delivery systems [51,54,59,60,61]. Figure 19 shows a subcutaneous (SC) administration of a polypeptide copolymer injectable hydrogel for protein delivery [7]. 

#### 2.2.2. DNA/Gene Delivery

Some mechanically improved IHs, such as sulfamethazine-containing multiblock polyurethanes (PUSMA) polymerized with hexamethylene diisocyanate (linker), are appropriate for the sustained release of cationic proteins due to their negative charge. The initial burst release between the cationic proteins and the anionic hydrogel network can be evaded. Due to the urethane-ester group in the anionic PUSMA copolymer, which possess good adhesive properties, the anionic PUSMA and DNA-loaded cationic polyplex IHs can adhere to the wounds and provide an aqueous matrix, which shows better wound healing results. The affinity-based sustained release of DNA-based cationic polyplex can be improved by the addition of heparin to the PUSMA complex [7,54,59].

#### 2.2.3. Vaccine Delivery

The conjugation of synthetic material with hyaluronic acid (HA) in IH synthesis, which undergoes a sol–gel transition phase in thermosensitive responses, has diverged from the current trends for successful vaccine delivery. The improved durability, degradability and mechanical toughness of the PCLA-b-PEG-b-PCLA co-polymer along with its spontaneous formation and flexibility contains a microporous network that encapsulates ovalbumin (OVA)-expressing DNA polyplexes. After its SC administration (hypodermic syringe), an in situ hydrogel reservoir is formed, which can migrate local antigen-presenting cells (APCs) to hydrogel surfaces and permeate microporous meshwork and released OVA-expressing DNA polyplexes and generate respective immune responses. The consequent antibody-dependent cell-mediated cytotoxicity may be able to progress into effective metastasis tumor suppression. This hybrid IH has been recognized as a prospective DNA-vaccine delivery platform for in vivo antigen-specific cancer immunotherapy [2,62]. Similarly, PCLA-b-PEG-b-PCLA-conjugated BSA is also an immune-responsive DNA-based IH vaccine that acts through humoral and cellular immune responses via dendritic cells (DCs) stimulation. After its administration through SC injections, microporous hydrogels attract DCs, which then travel into the skin and drain the lymph nodes. Hydrogel then gradually depletes under the skin, and encoded loaded DNA vaccines can be released and effectively taken up by the DCs, eliciting antigen-specific immune responses [2,7,61], see Figure 20 [63]. 

#### 2.2.4. Tissue Engineering

IHs and their applications in cartilage repair and regeneration have high clinical impact in biomedical engineering. During cartilage repair, the hydrogel implant is sustained within the defective site, as well as sustaining instantaneous weight-bearing, due to its mechanical strength and stiffness within minimally invasive surgery. Tissue engineering is considered as a rapid practice for combining scaffolds, cells and bioactive molecules for the maintenance, recovery and enhancement of tissue performance and regeneration. The biomaterial/polymer, with its predefined, porous biocompatible 3D structure, can fit within the anatomical site, where the cells can attach, fully grow and reorganize to produce entirely functional tissues. These biomaterials are biodegradable and bioabsorbable, provide appropriate mechanical strength, play a vital role in targeted delivery and allow the repair of damaged tissue without causing inflammatory cascades. Some of these biomaterials are triblock polymers of PLA–poly(ethylene oxide)–PLA, which is extensively used for nerve, orthopedic and soft tissue repair/regeneration applications. Similarly, PVA-grafted lactide oligomers are used as a cartilage substitute in tissue engineering. The other most commonly used hydrogel polymeric materials for scaffold fabrications are various proteoglycans, collagen, alginate-based substrates and chitosan [14,20].

#### 2.2.5. Regenerative Medicines

Currently, the reconstruction and regeneration of soft tissues are challenging tasks in biomedical engineering. A lack of soft tissues may be due to many reasons, including congenital trauma or disease, or oncology surgery—for instance, Poland syndrome or Parry-Romberg syndrome (congenital), which can progress into lipoatrophy. In the past, the replacement of autologous adipose tissues has been applied, where they may have variable graft resorption due to the lack of vascular system. IH-based regenerative medicines or tissue engineering, synthesized from natural and synthetic biomaterials, have replaced all these methods. For instance, Hemmerich et al. reported the reconstruction of small defects, in which hyaluronic acid-based IH scaffolds were associated with undifferentiated adipose-derived stem cells (ASCs) for adipose tissue regeneration. The natural polymer, hyaluronic acid, with its application for engendering adipose tissue in gels, exhibits adipogenic as well as angiogenic properties [25,64,65].

### 2.3. Therapeutic Applications

#### 2.3.1. Cancer Therapy

Systemic cytotoxicity is associated with the conventional chemotherapy of cancers. The localized non-invasive chemotherapeutics of IHs can overcome the associated toxicities and ensure drug/protein sustained release at tumor sites. In more detail, these IHs (natural and synthetic polymers) are responsive to certain stimuli, such as pH, temperature and immuno-sensitivity, which can be used for cancer chemotherapy/gene therapy or immune therapy [36,38,40,55,66]. Some of the commercially available synthetic IHs, such as TraceIT^®^ and SpaceOAR^®^, are therapeutically used for cancerous cell applications, which can protect healthy cells from radiotherapy-induced damage. TraceIT^®^ contains tissue markers and polyethylene glycol (PEG) hydrogel microparticles, with covalently bond iodine, which helps in the visualization and identification of cancerous tissue for up to three months using ultrasound, computed tomography (CT) and magnetic-resonance imaging (MRI). SpaceOAR^®^ hydrogel has also been fabricated to protect normal tissues from radiation during radiotherapy for carcinoma. Hence, improving the persistence and rigidity of natural IHs may have significant importance in the treatment of carcinomas and tumors ((Figure 21) [67] and (Figure 22) [36]).

#### 2.3.2. Wound Healings

As we know, common clinical traumas are associated with tumors, fractures and diabetes, which often go hand in hand with serious infections, including superbugs. Hydrogel dressings with excellent antibacterial properties, bioadhesiveness, good compatibility and biodegradability are preferred over conventional wound healing. They have been fabricated to control infections, improve angiogenesis and uphold angiogenesis. Hydrogels that are photothermal sensitive have the ability to convert NIR irradiation intensity time, photothermal-initiator concentration ratio, cycling time, etc. Increasing the temperature from >50 °C can inhibit bacterial growth effectively, while at 41–43 °C it accelerates wound closure and healing. Nanoparticles (NPs) of copper (GelMA/BACA–Cu) hydrogels possess good photothermal capability under NIR irradiations. After ten minutes of exposure to NIR, hyperthermia (>55 °C) can efficiently inhibit bacterial growth, while its Cu^2+^ combination can enhance its antibacterial efficacy. The wound closure rate was 95% in the NIR + GelMA/BACA–Cu NPs hydrogel group, in comparison with the control group, 79%. It also plays a major role in accelerating angiogenesis and encouraging the propagation of fibroblasts. Similarly, Chu et al. developed a NIR + Cu-carbon dots biomaterial in which the wound closure rate reached 96% (control = 62%) after treatments for 14 days, while its H&E staining showed more collagen depositions, neovascularization and re-epithelization in comparison with the control group. However, NIR-induced hyperthermia has short-term antibacterial efficacy, so when it is stopped, the remaining pathogens cannot be effectively inhibited. Therefore, in order to address these concerns issues, some studies have suggested the encapsulation of antibiotics in combination therapies [21,36,46], see Figure 23 [7] and Figure 24 [42].

#### 2.3.3. Bone Regeneration

According to Kim et al., alginate-based conjugating temperature-responsive poly(ε-caprolactone-co-lactide)-b-poly(ethylene glycol)-b-poly(ε-caprolactone-co-lactide) and O-phosphoryl-ethanolamine can be used as phosphorylation functional groups (PCLA-b-PEG-b-PCLA-/Alg) to obtain bioconjugate-injectable scaffold systems that have the ability to accelerate bone biomineralization [13] (Figure 25). When the temperature was elevated from room temperature to physiological temperature, these bioconjugates showed a sol–gel transition. In in vivo and in vitro situ biomineralization, the XRD analysis confirmed that such bioconjugate hydrogel could result in a reduction in crystalline hydroxyl apatite, and it also released BMP-2 (mitogenic factors) for more than three days. These BMP-2-containing PCLA-b-PEG-b-PCLA/Alg bioconjugate hydrogels also possess calcium deposition at the eroded sites, which suggests mineralization and bone regeneration; see Figure 25 [13,52].

### 2.4. Biodegradable Hydrogel Injectables That Are Undergoing Clinical Trials 

Table 3 shows all hydrogels that are under clinical trial (Phase I, II, II or IV). The diversity of these crosslinking polymers/biomaterials (natural/synthetic) used in the synthesis of hydrogel scaffolds reduces their regulatory classification and results in approval challenges. They were broadly classified under the “device” category, which according to Section 201 (g) of the FD&C Act, covers “any product which does not achieve its primary intended purposes through chemical action within or on the body”. Furthermore, other than a few exceptions, the majority of these hydrogel-based products are required to undergo an additional FDA review of a 510 (k) pre-market notification submission for approval and legal marketing rights in the US. Hydrogel scaffolds that encapsulate a drug, or drug-secreting cells, are considered as a combination product, and thus their regulatory approval takes up to 7 to 12 years or more, which further restricts their commercial availability [25].

### 2.5. FDA-Approved Hydrogel Formulations

Table 4 contains some of the FDA-approved commercially available in situ IHs. The brand name, gelation mechanism, polymer type, injection type, indications and FDA-approved status are noted.

## 3. Future Prospects

### 3.1. Limitations and Outcomes/Overcomes

Regarding their current prospective and ongoing research, hydrogel formulations have some limitations in their applications, clinical practices and sustainability. Many hydrogel systems (natural/synthetic), such as thermosensitive hydrogels, as mentioned, are free-flowing sols at a low temperature, while upon raising to body temperature (physiological temperature), they are converted to a stable visco-elastic gel phase, such as poly (phosphazene), pluronic and poly (*N*-isopropyl acrylamide). However, their biomedical uses are restricted due to their limited non-biodegradable and biocompatible characteristics, as shown in Table 5. However, poly (ester)-based copolymers are a good alternative for overcoming these limitations, but they still require more extensive research. Moreover, PEG and poly(ester)-based hydrogels (local drug-carriers) are not effective for prolonged therapeutics. Their oral and nasal route administrations are inappropriate, though they have been approved by the FDA for in vivo implantation. Similarly, injectable hydrogels used for proteins and peptide-controlled delivery, chemical interactions, structure compatibilities and burst release (when charged proteins are added to uncharged formulations) are also facing future challenges [13].

In enzymatic stimuli responsiveness, fragile molecules, such as proteins, cells and drugs, are degraded or denatured by the use of cross-linker monomers and toxic catalysts. Homogenous encapsulation, loading and pre-mature release are also associated limitations and concerns [4].

### 3.2. Injectable Formulation Challenges

Here, we describe the ongoing prospective of a targeted drug delivery system that improves the therapeutic regimen and enables target-specific delivery. In addition, self-assembled nanocarriers and their active targeting receptors/antigen overexpression in tumor cells are prospective major therapeutic approaches.

IHs, along with their specific applications, which have been already discussed, have some common notable concerns and require further exploration. The compatibility of the fragile molecules or cells with hydrogel crosslinkings and proliferation maintenance within a healthier environment are of the utmost importance, while they are essential for protecting DNA, peptides, proteins and oligonucleotides from enzymatic degradation or denaturation. Cytotoxicity and inflammatory responses need to be avoided by under-standing their interactions with the cellular and surrounding tissues. Low reproducibility; poorly defined structures; employed system considerations, including the gelation mechanism, rate kinetics, viscosity during the injection time; mechanical sturdiness after gelation; the degradation period; and release profile of bioactive factors should be properly taken in account [31]. It should be noted that these IHs must have application-specific design measurements related to their chemical–physical crosslinking and biological computability, which should be specific and responsive to certain medical conditions or pathogenesis [21,57].

#### 3.2.1. Mechanical Robustness

The maintenance of a low viscosity and sustainability of sufficient elasticity in situ for repetitive load and volume is a major design challenge [32]. As these hydrogels are administered through a needle and syringe, in situ gelation and repetitive doses are of great concern. Shear thinning polymers, such as hyaluronic acid, which are currently used as a dermal filler, are being replaced by an injectable for cartilage replacement [2,11,15]. Thus, the polymer molecular weight, chains, extent of crosslinking and crosslinking mechanism and viscosity have a linear relationship with the molecular weight, while the elastic modulus has an inverse relation [25,57].

#### 3.2.2. Loading and Release of Therapeutic Agents

The loading and release of therapeutic agents, such as small drug molecules, macromolecules (peptides, proteins, nucleic acid, etc.) or living cells, can be conveyed to the surrounding environment through IH carriers, which have respective physico-chemical characteristics. Their effective release is determined by the cargo-gel size, affinity and interactions. Currently, IHs in clinic practice include microparticle depot systems (small molecules or biologics). The most commonly available depot formulations are lidocaine (anesthetic agent), with a number of hyaluronic acid hydrogel injectables that have been approved for facial corrections. Hydrogel meshes disturb lidocaine release due to its rapid release from these gels. Similarly, the elution of other drugs from hydrogel formulations for the sustained release of proteins/drugs in wound dressings is vital [25]. In such formulations, the hydrogel mesh size should be decreased via physical/chemical crosslinking to agitate the solute elution or increase its affinity to enhance its retention time [65].

#### 3.2.3. Hydrogel Bioactivity

The bulk hydrogel material must penetrate, alter and degrade bulk hydrogel materials for tissue regeneration purposes. For hydrogel bioactivity, it is essential that cells or growth factors stick to adhesive natural or synthetic materials such as hyaluronic acid, fibrin or gelatin. For instance, in some ongoing clinical trials, e.g., NCT04115345, NCT04115345 and NCT00981006, the hydrogel bioactivity of gelatin is used with cells and/growth factors. These bioactive hydrogels expedite kidney or myocardium tissue injury healing by generating a suitable cell-adhesive microenvironment. Some non-adhesive polymers, such as PEG and polyacrylamide, must be amended chemically with adhesive ligands to assist cell penetration and attachment. Conversely, some commercially available hydrogels, e.g., TraceIT^®^, which is used for targeting tumor margins, have the design limitation of rapid biodegradation. Thus, it is essential to use or substitute a polymer hydrogel that degrades gradually over weeks to months [1,10,12,25,64].

#### 3.2.4. Immunological Compatibility

Provoking immunological responses to IH biomaterials has been a considerable area of research in recent decades. Thus, in injectable hydrogel formulations, it is imperative to minimize all kinds of immune responses during in situ gel transitions [74,75]. Immunological responses are considered as the worst consequences of biomaterial injection, insertion and implantation, including inflammatory cascades, fibrosis and hypersensitivity reactions. These responses are considered harmful in their own prospects, associated physico-chemical shifts and responsiveness (i.e., changes in the local pH or temperature), which can affect the hydrogel material performance, function and further application [76]. Thus, minimalizing IH-associated host-immune responses is a critical biological design parameter. Cell-based drug delivery systems (erythrocytes, leukocytes, platelets, cancer cells and hepatic cell membrane biomimetic nanosystem fabrications) can be utilized for immunocompatibility, immune escaping, prolonged release and sustained release [77,78,79,80,81,82,83,84,85,86,87,88,89]. Some hydrogels have been used in these cell-based fabricated biomimetic nanosystems [90].

#### 3.2.5. Technological Challenges

Technological challenges, including chemistry, good manufacturing practices (GMP), controlled and well-defined regulatory guidelines, practical adoptability and high prices are major hurdles for successful clinical translations of hydrogel-based delivery systems. It is estimated that hydrogel fabrication system development and fabrication costs through clinical translation range from USD 50 million up to USD 800 million.

#### 3.2.6. Scale-Up Strategies and GMP Processes

As most hydrogel systems are generated and synthesized at a small pilot-plant scale at pre-clinical stages, current good manufacturing practices (cGMP) for the clinical translation and integration of biomaterial-based hydrogels in large-scaled systems and their compatibility are required. Robustness, batch-to-batch variations, safety profiles, reproducibility and proficiency are predictable when executed at a large scale. Moreover, the high water content of hydrogels makes the synthesis, fabrication, storage, sustainability, sterilization and all relevant optimization processes even more demanding.

#### 3.2.7. Regulatory Approvals

As we mentioned earlier, regulatory affairs and FDA approval are time-consuming processes and take years—from laboratory synthesis to market launch and surveillance. The diversity in injectable hydrogel scaffolds and the assortment of employed crosslinking polymers and biomaterials render their regulatory classification and approval challenging [25].

## 4. Conclusions

In conclusion, bearing in mind all the aspects of IHs, their biomedical applications still need further exploration and extensive research. It seems that their innovative prospects can give rise to new information and revolutions, especially in biomedical engineering, tissue regeneration, drug/protein/gene delivery, cancer chemotherapies, wound dressings, implants, superbugs targeting, etc. The current prospective hydrogel research; FDA-approved formulations; clinical trials; biomedical applications; molecular-level studies; crosslinking modules; and natural, synthetic and natural–synthetic hybrid synthesis are widening hydrogels application and will diversify their therapeutic use in other health ailments and complications. Thus, highlighting all these limitations, the associated demerits and design challenges in injectable formulations will pave the way for and suggest future prospective research into efficient and more inventive applications.

## Figures and Tables

**Figure 1 pharmaceuticals-15-00371-f001:**
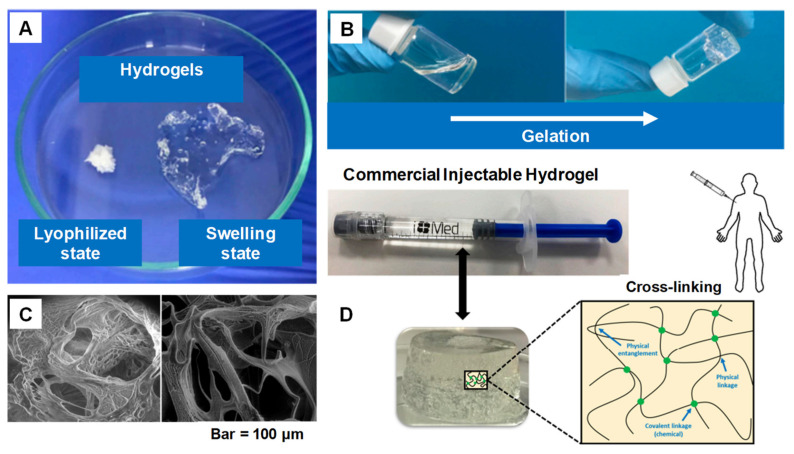
(**A**) Swelling ability of hydrogels from lyophilized state. (**B**) Example of a gelation time measurement by inverted vial method. (**C**) Cross-section SEM images of freeze-dried injectable hyaluronic acid hydrogel. (**D**) Physical interactions and chemical linkages of the chemical structure of an injectable hydrogel. Adapted with permission from ref. [9]. Copyright © 2021 *Polymer* MDPI.

**Figure 2 pharmaceuticals-15-00371-f002:**
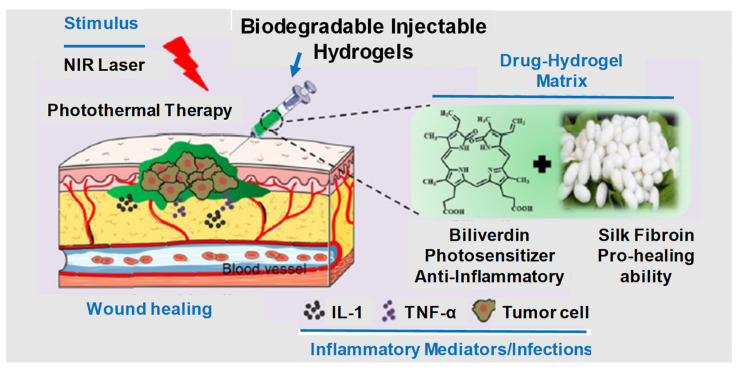
Graphical illustration depicting the design and fabrication of injectable hydrogel for photothermal antitumor therapy and following wound repair and regeneration. Adapted with permission from ref. [10]. Copyright © 2020 *Theranostics*.

**Figure 3 pharmaceuticals-15-00371-f003:**
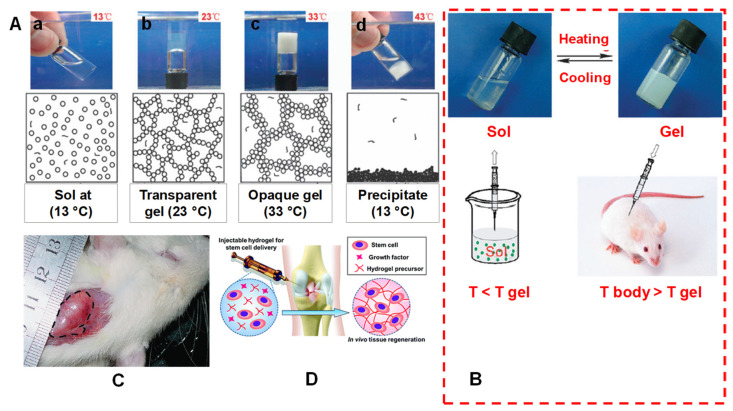
(**A**) Mechanism of a spontaneous thermos gelling of the appropriate block copolymers in water via the formation of a ‘‘micelle-network’’. For simplicity, a micelle is denoted by a circle, although a micelle owns the core-corona structure and is deformable. (**a**) The aqueous system takes on a sol-like suspension at a low temperature; (**b**) the micelles are aggregated into a percolated micelle network in which each micelle is still intact, but micelle aggregation occurs due to the hydrophobic interaction between micelles, and the solvent loses flowability, leading to the so-called sol–gel transition; (**c**) the micelle network is coarsened until the mesh size is in the order of the visible light wavelength, and the gel is thus opaque; (**d**) the micelle structure is destroyed due to over-hydrophobicity of the sample at higher temperatures, eventually leading to macroscopic precipitation. (**B**) A schematic illustration of an injectable hydrogel system exemplified by a physical thermos-gelling material. T gel is the sol–gel transition temperature. The polymers could be dissolved in water to form a sol at low temperatures. Bioactive molecules or cells indicated by the dots in the lower-left image can be incorporated by simple mixing with sols. The sols are injectable, and in situ gelling takes place after injection if the gelling temperature is lower than the body temperature T body. As a result, the encapsulation of drugs or cells and the implantation of biomaterial are carried out with minimal surgical invasiveness. (**C**) A global observation of a physical gel formed underneath the skin of a rat. The image was taken 21 days after subcutaneous injection of an aqueous solution of PLGA–PEG–PLGA copolymer. The gel region is emphasized by the dashed line. Adapted with permission from ref. [11]. Copyright © 2008 *Chemical Society Review*. (**D**) Injectable hydrogel for stem cell therapy. Adapted with permission from ref. [15]. Copyright © 2019 *The Royal Society of Chemistry*.

**Figure 4 pharmaceuticals-15-00371-f004:**
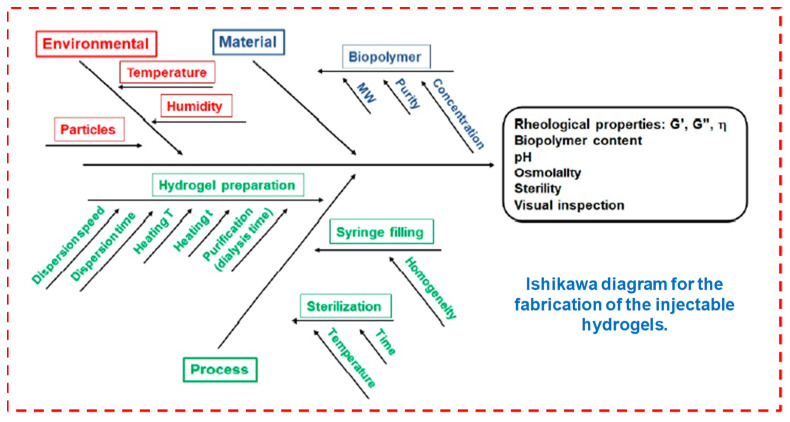
Fabrication of injectable hydrogels. Reprinted with permission from ref. [9]. Copyright © 2021 *Polymer* MDPI.

**Figure 5 pharmaceuticals-15-00371-f005:**
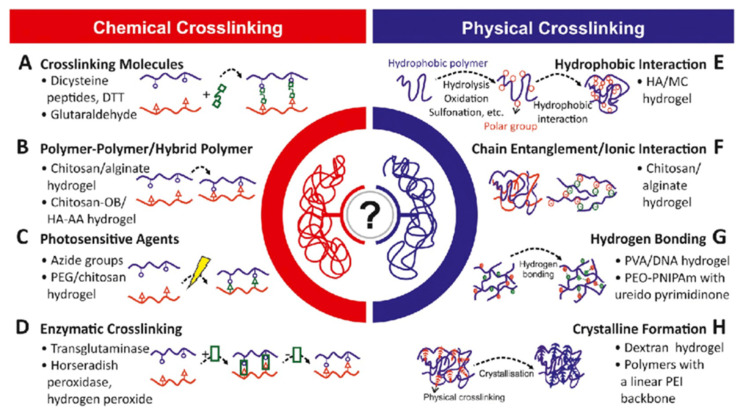
Polymers and crosslinking physico-chemistry. Reprinted with permission from ref. [22]. Copyright © 2019 *Biotechnology advances* Elsevier.

**Figure 6 pharmaceuticals-15-00371-f006:**
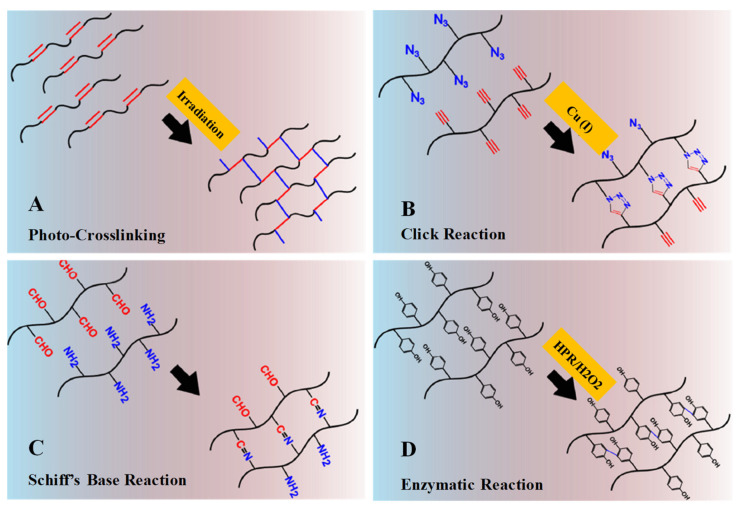
Schematic illustrations of chemical crosslinking mechanism: (**A**) Gelation by photo-crosslinking reaction of vinyl groups bearing polymers. (**B**) Gelation by alkyne-azide click reaction with Cu (I) as catalyst. (**C**) Gelation by Schiff’s base reaction between aldehyde and amine groups. (**D**) Gelation by enzymatic reaction with horseradish peroxidase and hydrogen peroxide as catalyst system. Adapted with permission from ref. [8]. Copyright © 2015 *European Polymer Journal*.

**Figure 7 pharmaceuticals-15-00371-f007:**
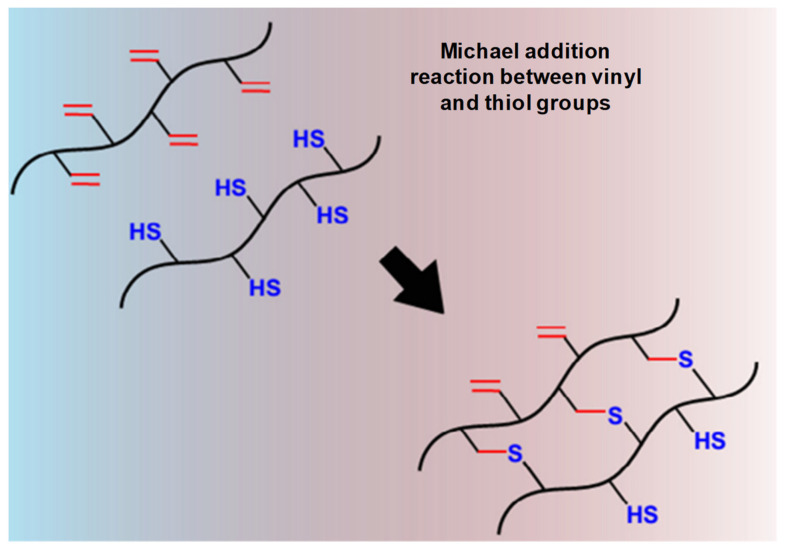
Schematic thiol-based Michael reaction between vinyl and thiol groups mechanism. Adapted with permission from ref. [8]. Copyright © 2015 *European Polymer Journal*.

**Figure 8 pharmaceuticals-15-00371-f008:**
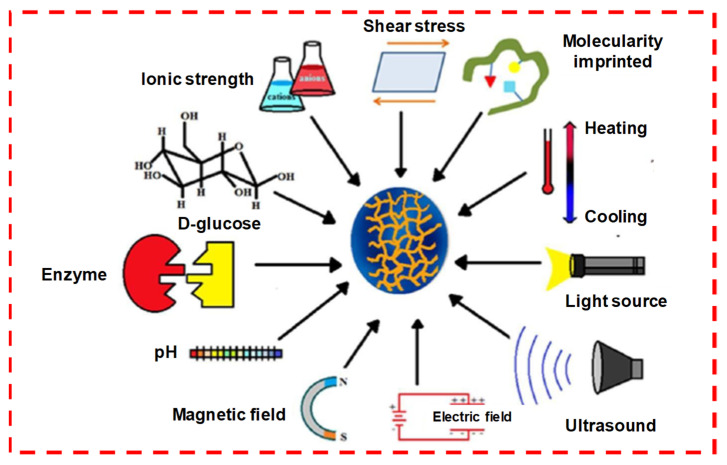
Physically crosslinking stimuli sensitive to hydrogels. Reprinted with permission from ref. [30]. Copyright © 2019 *European polymer journal*, MDPI.

**Figure 9 pharmaceuticals-15-00371-f009:**
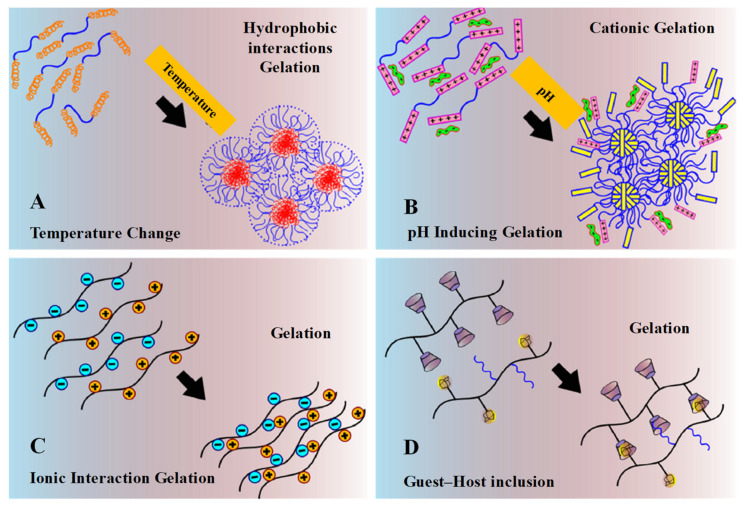
Schematic physical crosslinking mechanism. (**A**) Gelation driven by shifting of hydrophobic interaction under the change of temperature. (**B**) Gelation driven by pH-inducing protonation–deprotonation transition of cationic hydrogel. (**C**) Gelation by ionic interaction. (**D**) Gelation by guest–host inclusion complexation between cyclodextrin groups and adamantine groups or PEG chains. Reprinted with permission from ref. [8]. Copyright © 2015 *European Polymer Journal*.

**Figure 10 pharmaceuticals-15-00371-f010:**
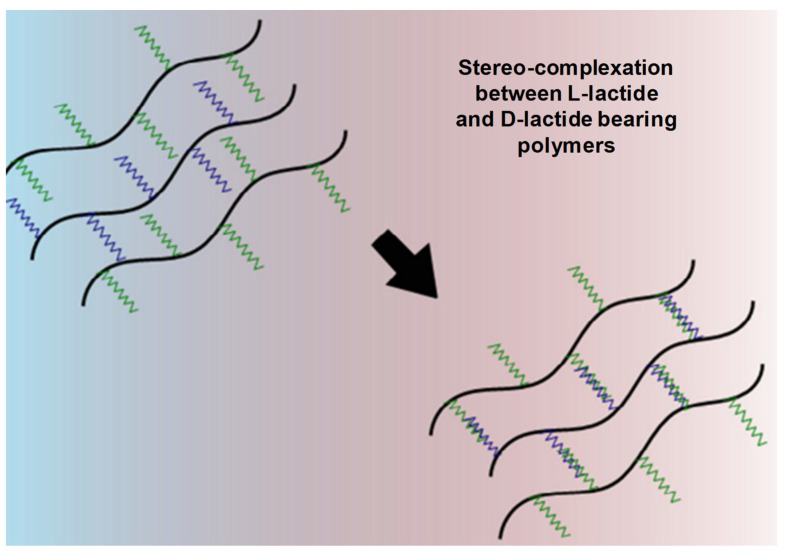
Mechanism of gelation by stereo-complexation between D-lactide- and L-lactide-bearing polymers. Adapted with permission from ref. [8]. Copyright © 2015 *European Polymer Journal*.

**Figure 11 pharmaceuticals-15-00371-f011:**
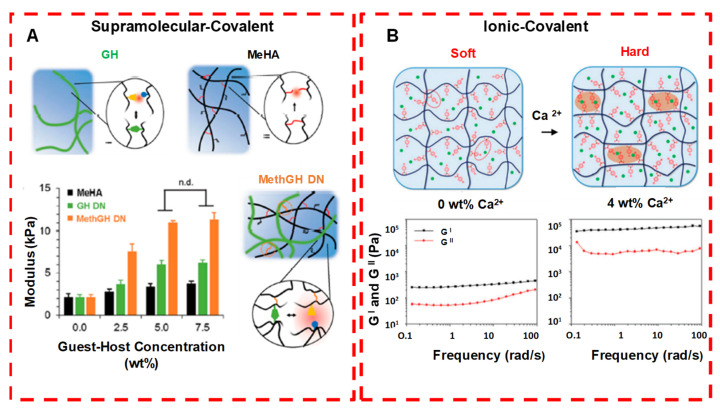
Crosslinking strategies for forming injectable double network hydrogels. (**A**) Supramolecular and covalently crosslinked hyaluronic acid hydrogels: by increasing the guest−host concentration or the chemical cross-linker concentration, gel modulus is enhanced. (**B**) Ionically and covalently crosslinked PVA−CPBA hydrogels: by increasing the concentration of Ca^2+^ ions, gel modulus is enhanced. n.d. means no significant difference. Adapted with permission from ref. [34]. Copyright © 2020 *ACS Applied Polymer material*.

**Figure 12 pharmaceuticals-15-00371-f012:**
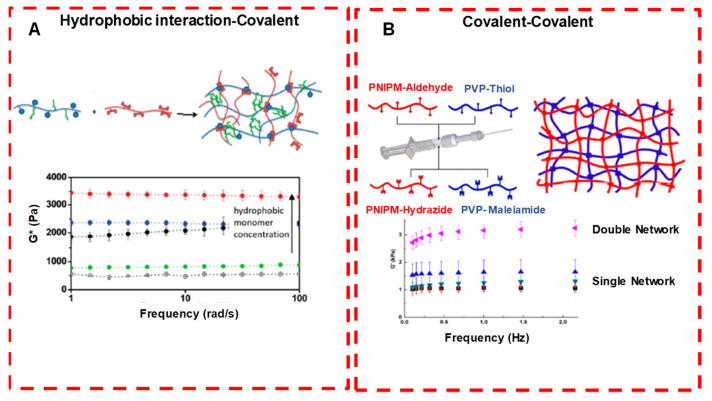
Crosslinking strategies for forming injectable double network hydrogels. (**A** Hydrophobically associated and covalently crosslinked POEGMA hydrogels: by increasing the concentration of the hydrophobic co-monomer, gel modulus is increased. (**B**) Dual covalently crosslinked hydrogels composed of aldehyde/hydrazide functionalized PNIPAM and thiol/maleimide functionalized PVP: the presence of both kinetically orthogonal crosslinking groups results in enhanced mechanics compared with either single network alone. Adapted with permission from ref. [34]. Copyright © 2020 *ACS Applied Polymer material*.

**Figure 13 pharmaceuticals-15-00371-f013:**
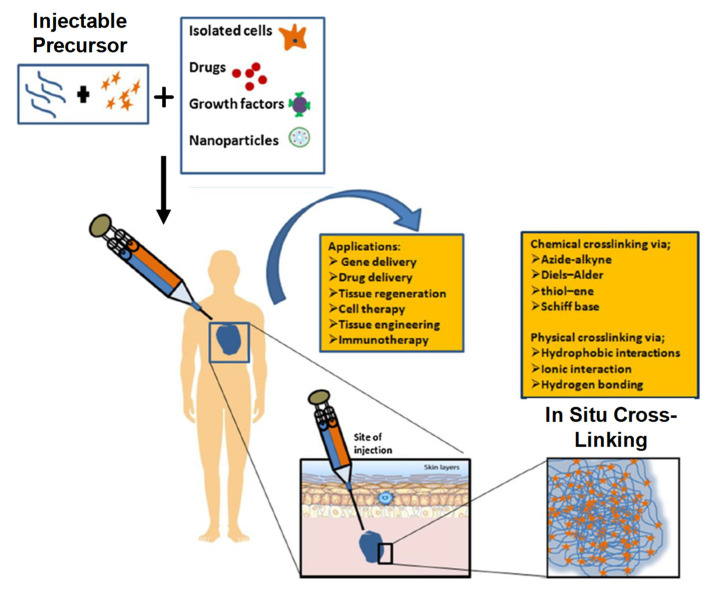
Mechanism of action of injectable hydrogel systems. Adapted with permission from ref. [36]. Copyright © 2017 *International Journal of Biological Macromolecules*, Elsevier B.V.

**Figure 14 pharmaceuticals-15-00371-f014:**
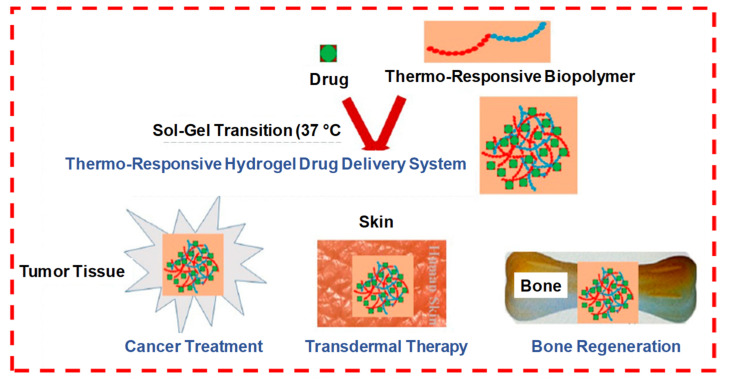
The formation of a drug-loaded biopolymer-based thermo-responsive hydrogel system via in situ gel formation and its biomedical applications, including cancer treatment, transdermal therapy and bone regeneration. Adapted with permission from ref. [5]. Copyright © 2020 *Polymers* MDPI.

**Figure 15 pharmaceuticals-15-00371-f015:**
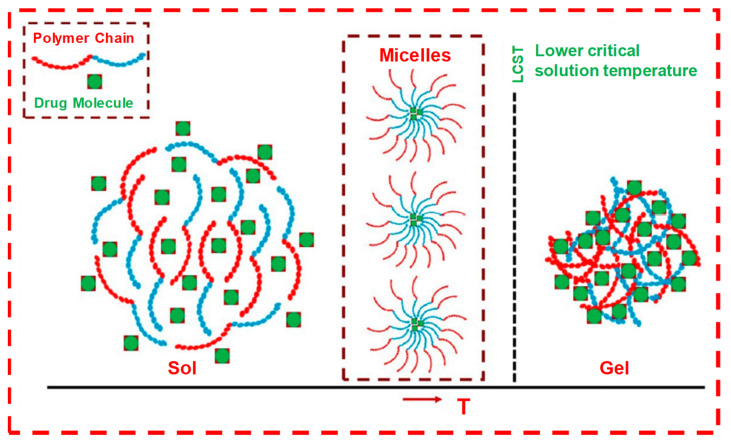
The sol–gel transition of LCST-type thermo-responsive polymer-based drug delivery system. Formulation over LCST changes to hydrogel from the solution state. LCST-type thermo-responsive polymer in solution forms micelles at low concentration, which further aggregates at high polymer concentration to form gel at a temperature ≥LCST. Adapted with permission from ref. [5]. Copyright © 2021 *Polymer* MDPI.

**Figure 16 pharmaceuticals-15-00371-f016:**
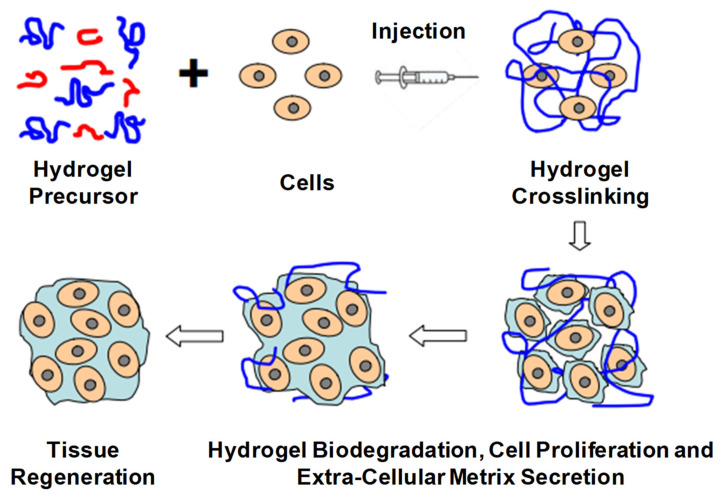
Schematic illustration of biodegradable injectable hydrogel for tissue regeneration approaches. Cells are isolated from a small biopsy, expanded in vitro and encapsulated in hydrogel precursors, which are subsequently transplanted into the patient by injection using a needle. Hydrogels provide initial structural support and retain cells in the defective area for cell growth, metabolism and new extracellular matrix (ECM) synthesis. The hydrogel is readily degradable when the cells secrete ECM. This strategy enables the clinician to transplant the cell, growth factor and hydrogel combination in a minimally invasive manner. Adapted with permission from ref. [6]. Copyright © 2010 *Material* by MDPI.

**Figure 17 pharmaceuticals-15-00371-f017:**
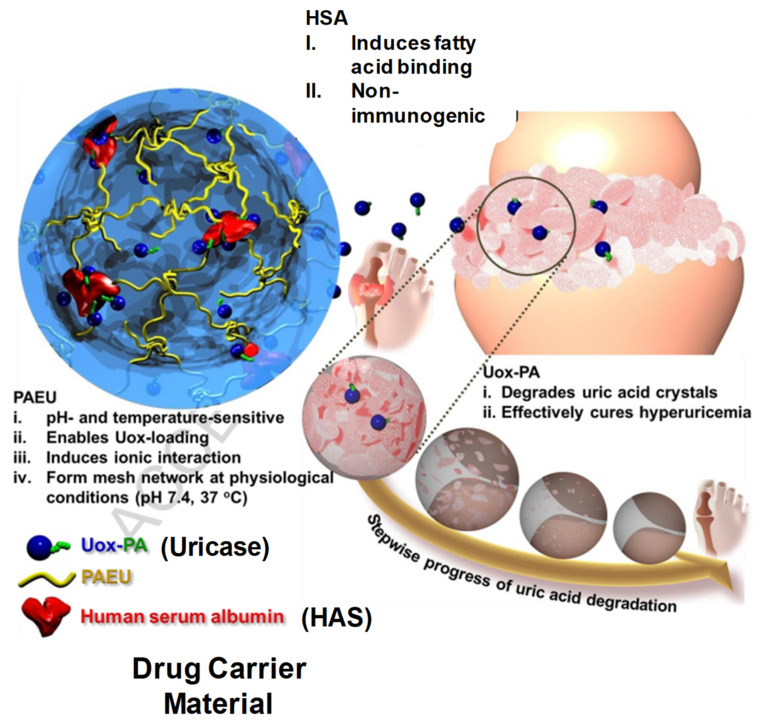
Schematic illustration of Uox-loaded PEG-PAEU/HSA hydrogels and their controlled release of Uox for degradation of uric acid crystals in hyperuricemia mice models. Reprinted with permission from ref. [57]. Copyright © 2017 from *Journal of Control Release*, Elsevier.

**Figure 18 pharmaceuticals-15-00371-f018:**
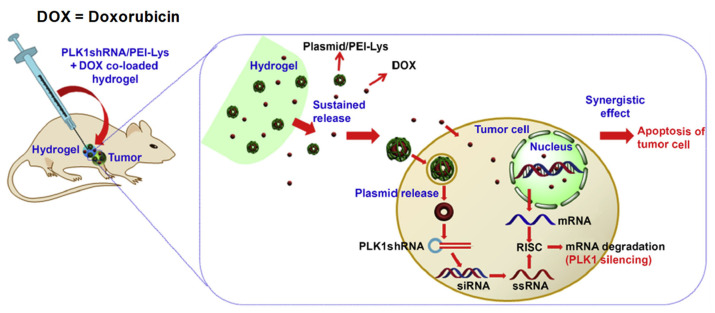
Schematic illustration for the synergistic effect of PLK1shRNA/PEI-Lys and DOX co-loaded hydrogel on a tumor in nude mice. Reprinted with permission from ref. [58]. Copyright © 2014 from *Biomaterial*, Elsevier.

**Figure 19 pharmaceuticals-15-00371-f019:**
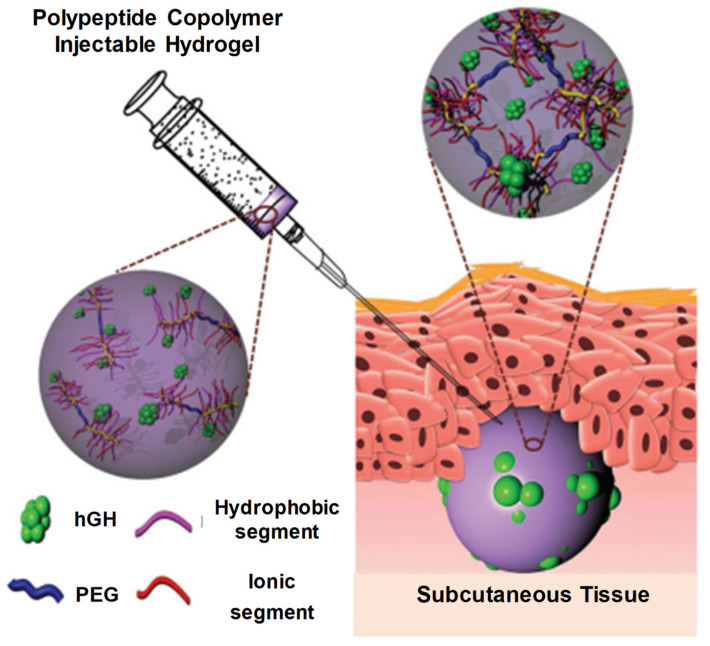
Schematic diagram showing injectable and hGH-loaded PNLG-co-PBLG-b-PEG-b-PBLG-co-PNLG hydrogels (a polypeptide copolymer) and their application for protein delivery. The polypeptide copolymers exist as a sol in the syringe and are transformed to a gel after subcutaneous administration. Reprinted with permission from ref. [7]. Copyright © 2021 *Journal of Control Release*, Elsevier.

**Figure 20 pharmaceuticals-15-00371-f020:**
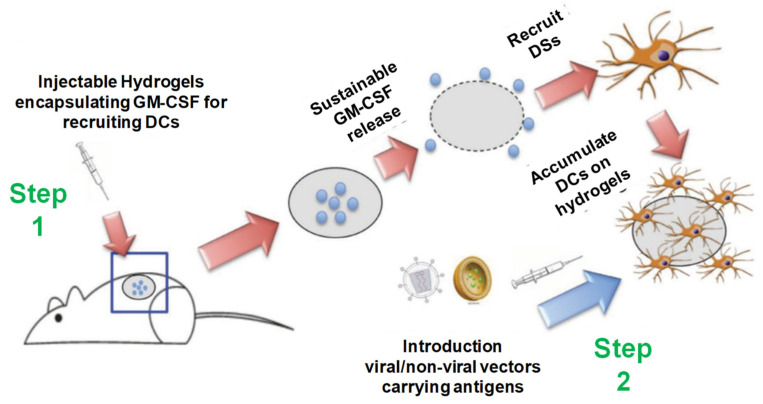
Injectable thermosensitive hydrogels for cancer vaccines. Step 1: Sustainable release of GM-CSF from the injectable thermosensitive mPEG- PLGA hydrogels recruits host dendritic cells (DCs) to the site of administration. Step 2: Viral or nonviral vectors carrying immunogens can be delivered in situ to the resident DCs in hydrogels to enhance antigen uptake efficiency, thereby improving anticancer immunity. Reprinted with permission from ref. [63]. Copyright © 2014 *Biomacromolecules*, American Chemical Society.

**Figure 21 pharmaceuticals-15-00371-f021:**
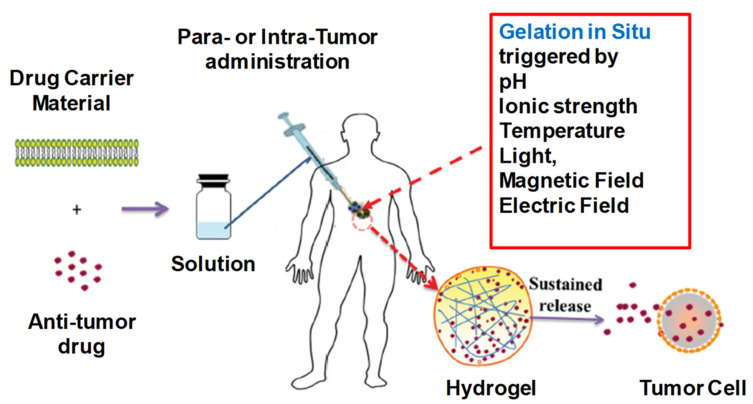
Depiction of in situ gel formation process and its sustained release of drugs from the hydrogel into tumor cells. Reprinted with permission from ref. [67]. Copyright © 2020 from *Drug delivery*, Taylor and Francis.

**Figure 22 pharmaceuticals-15-00371-f022:**
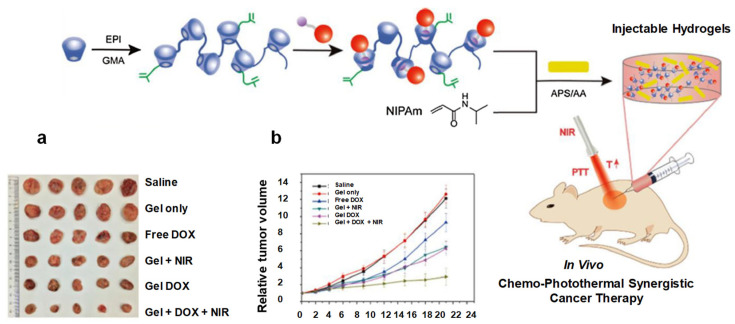
Schematic diagram showing hydrogel formation. (**a**) Photograph of excised tumors after mice were euthanized on the 21st day. (**b**) Changes in relative tumor volume. Reprinted with permission from ref. [36]. Copyright © 2020 Copyright © 2017, *International Journal of Biological Macromolecules*, Elsevier.

**Figure 23 pharmaceuticals-15-00371-f023:**
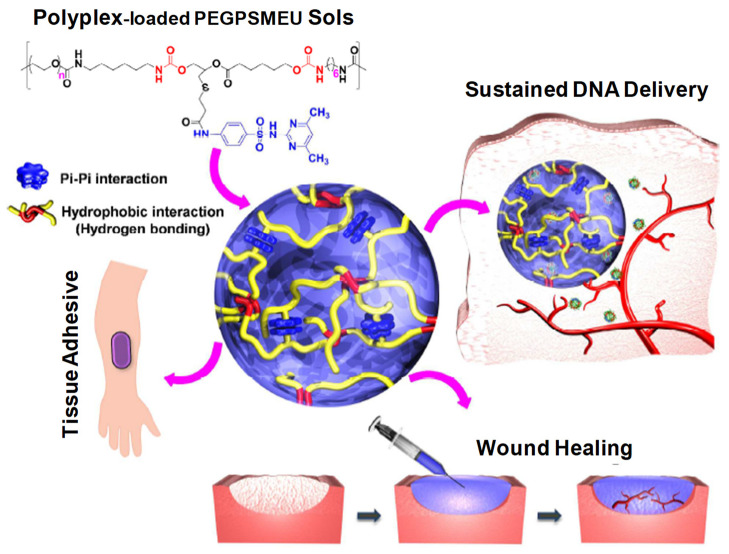
Schematic concept of sol-to-gel phase transition of polyplex-loaded PEGPSMEU copolymers sols, subcutaneous administration, controlled released via diffusion, effective absorption onto skin and wound healing. Reprinted with permission from ref. [7]. Copyright © 2020 from *Journal of Controlled Release*, Elsevier.

**Figure 24 pharmaceuticals-15-00371-f024:**
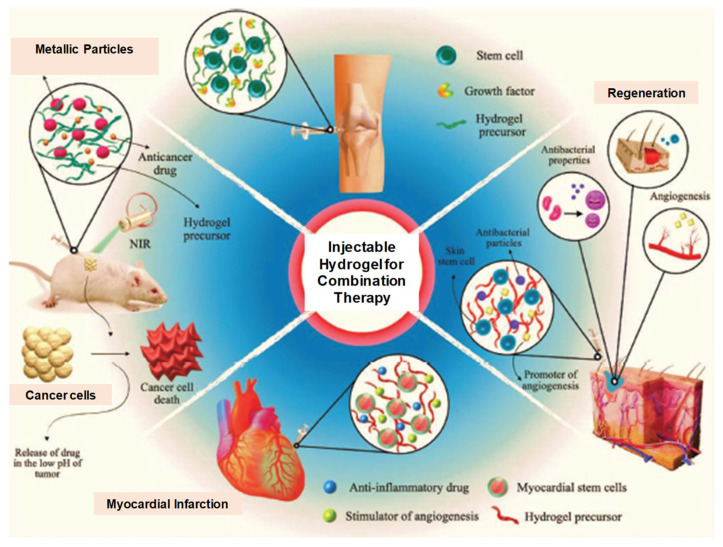
Illustration of the potential application of multifunctional IHs for cancer multi-therapy as well as regeneration after tissue damage. Reprinted with permission from ref. [42]. Copyright © 2020 *Advance Health Care Materials*, Wiley-VCH GmbH.

**Figure 25 pharmaceuticals-15-00371-f025:**
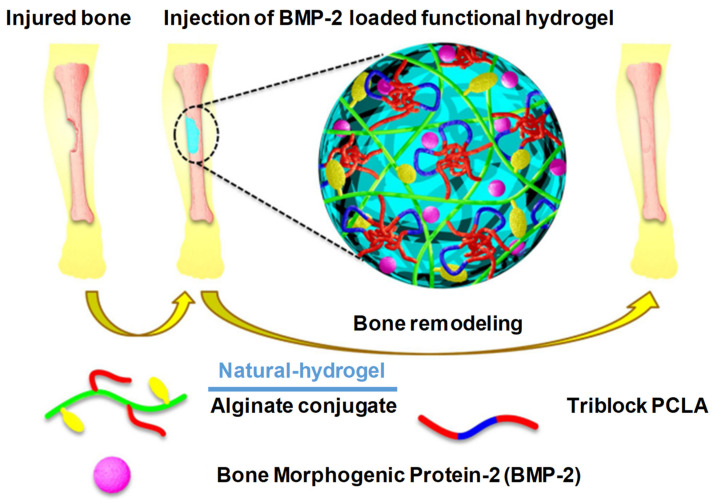
Schematic illustration of recruitment of host cells into PCLA-b-PEG-b-PCLA/BSA hydrogels and subsequent activation in response to DNA vaccine-bearing polyplexes released from these hydrogels. Reprinted with permission from ref. [7]. Copyright © 2020 from *Journal of Controlled Release*, Elsevier.

**Table 1 pharmaceuticals-15-00371-t001:** Classification of hydrogels on the basis of source, structure, crosslinking, charge and biodegradation.

Classification of Hydrogel		Types with Examples		
Source	Natural Agarose, Alginate, Chitosan, Collagen I, Fibrin, Gelatin, Hyaluronic acid, Matrigel	Synthetic Gelatin methacryloyl, Pluronic, Polyethylene glycol (PEG), Polyamides, Poly (acrylic acid)		
Structure	Inter-penetrating network	Co-polymer network	Homopolymer network	Double network
Crosslinking method	Chemical crosslinking		Physical crosslinking	
Charge	Anionic	Cationic	Amphoteric	Non-ionic
Biodegradable method	Non-biodegradable: Poly(2-hydroxyethyl methacrylate, Trimethylolpropane trimethacrylate),		Biodegradable natural: Collagen/Gelatin, Chitosan, Hyaluronic acid, Chondroitin sulfate, Alginate, Agar/Agarose, Fibrin Synthetic: Polyethylene glycol, polyethylene oxide, poly-vinyl alcohol, Poly(aldehyde guluronate), Polyanhydrides	

**Table 2 pharmaceuticals-15-00371-t002:** Hydrogel-based commercially available dosage forms and their applications.

	Brand/Commercial Product	Polymer	Active Constituents	Dosage Form	Application	Manufacturer
Hydrogel-Based Oral Dosage Form	Buccastem^®^ M	Povidone K30, xanthan gum, locust bean gum	Prochlorperazine maleate	Tablet	Nausea and vomiting in migraine	Alliance Pharmaceuticals, Chippenham, UK
	Biotene^®^	Carbomer and hydroxyethyl cellulose	Nil	Gel	Oral moisturizing agent in dry mouth	Glaxo SmithKline, London, UK
	Gengigel^®^	Hyaluronan	Nil	Gel	Mouth and gum care—oral ulcers	Pharmaniaga Berhad, Selangor, Malaysia
	Hydrogel 15%	Carbomer in ozonized sunflower oil	Ozone	Gel	Oral health	Honest 03, Dimondale, Michigan, USA
	Lubrajel™ BA	Glyceryl acrylate and glyceryl polyacrylate	Nil	Gel	Oral moisturizing agent	Ashland Global Specialty Chemicals Inc., North Calorina, USA
	Nicorette^®^	Hydroxypropyl methylcellulose	Nicotine	Chewing gum	Smoking cessation	Glaxo SmithKline, London, UK
	Nicotinell^®^	Xanthan gum and gelatin	Nicotine	Chewing gum	Smoking cessation	Glaxo SmithKline, London, UK
	Zilactin-B Gel^®^	Hydroxypropyl cellulose	Benzocaine	Gel	Local anesthetic in minor oral problems	Blairex laboratories Inc, Indiana, USA
	Zuplenz TM	Polyethylene glycol 1000, polyvinyl alcohol and rice starch	Ondansetron	Soluble oral film	Chemotherapy, radiation, surgery-induced nausea and vomiting	Galena Bipharma Inc., Portland, USA
Hydrogel-Based Ocular Dosage form Dosage Form	Biofinity^®^	Comifilcon A	Silicone hydrogel	Ocular	Continuous wear up to 7 days, corrects near sightedness and far sightedness	Cooper Vision, California, USA
	Air Optix^®^ night and day aqua	Lotrafilcon-A	Fluoro-silicone hydrogel	Contact lenses	Continuous wear up to 7 days, corrects near sightedness and far sightedness	Alcon, Texas, USA
	Retisert^®^	Silicone elastomer and polyvinyl alcohol membrane	Fluocinolone acetonide	Intraocular implant	Deliver long-term control of inflammation	Bausch and Lomb, New York, USA
	Lacrisert^®^	Hydroxypropyl cellulose	Nil	Ophthalmic insert	Moderate to severe dry eyes	Bausch and Lomb, New York, USA
	Systane^®^	Propylene glycol	Aminomethylpropanol	Ocular lubricant	For use as a lubricant to prevent further irritation or to relieve dryness of the eye	Alcon, Texas, USA
	Restasis^®^	carbomer copolymer Type A	Cyclosporine	Insert into eye	Indicated to increase tear production	Allergan, California, USA
	Proclear^®^ (Omafilcon B)	2-Hydroxy-ethylmethacrylate and 2-methacryloxyethyl phosphorylcholine crosslinked with ethylene glycol dimethacrylate	Nil	Contact lenses	Indicated for daily wear for the correction of visual acuity	Cooper Vision, California, USA
	Clintas Hydrate^®^	Carbomer	Nil	Eye	Lubricating eye gel for occasional dry eye discomfort	Altacor, Cambridge, UK
	Dailies^®^ AquaComfort	Nelfilcon A polymer (polyvinyl alcohol partially acetalized with *N*-formylmethyl acrylamide)	Nil	Contact lenses	Optical correction of refractive ametropia	Ciba vision, Atlanata, Georgia
	Systane^®^ gel drops	Polyethylene glycol 400, propylene glycol	Nil	Eye instillation	For the temporary relief of burning and irritation due to dryness of the eye	Alcon, Texas, USA
	Hylo^®^ gel	Sodium hyaluronate, citrate buffer, sorbitol	Nil	Eye instillation	Long lasting dry eye relief	Candorvision, Quebec, Canada,
	Iluvien^®^	Polyvinyl alcohol, and silicone adhesives	Fluocinolone acetonide	Intravitreal implant	Treatment of diabetic macular edema	Alimera Sciences, Alpharetta, Georgia
	Yutiq™	Polyvinyl alcohol	Fluocinolone acetonide	Intravitreal implant	Treatment of chronic non-infectious uveitis affecting the posterior segment of the eye	EyePoint Pharmaceuticals Inc, Massachusetts, USA
	Ozurdex^®^	Poly (D,L-lactide-co-glycolid)	Dexamethasone	Intravitreal implant	Macular edema, non-infectious uveitis	Allergan, California, USA
Hydrogel-Based Wound Dressing Dosage Form	Helix3-cm^®^	Type 1 native bovine collagen	Nil	Dermal gauze pad	Management of burns, sores, blisters, ulcers and other wounds	Amerx Health Care Corp, Florida, USA
	3M™ Tegaderm™ hydrogel wound filler	Propylene glycol, a hydrocolloid dressing	Nil	Dermal wound filler	Low to moderate draining wounds, partial and full-thickness dermal ulcers	3M Health Care Ltd., Minnesota, USA
	AquaSite^®^ amorphous hydrogel dressing	Glycerin-based hydrocolloid dressing	Nil	Wound dressing	Provide moist heat healing environment and autolytic debridement	Integra Life Science Corp, New Jersey, USA
	Algicell^®^ Ag calcium alginate dressing with antimicrobial silver	Calcium alginate ionic silver	Silver	Infective wound dressing	Effective against a broad range of bacteria and more absorption of drainage	Integra Life Science Corp, New Jersey, USA
	INTRASITE^®^ gel hydrogel wound dressing	Modified carboxymethyl cellulose, propylene glycol	Nil	Necrotic wound dressing	Re-hydrates necrotic tissue, facilitating autolytic debridement minor burns, superficial lacerations, cuts and abrasions	Smith &Nephew Healthcare Limited, Watford, UK
	Microcyn^®^ skin and wound hydrogel	Hypochlorous acid	Nil	Wound dressing	All types of chronic and acute wounds and all types of burns	Microsafe Group, Adelaide, Australia
	Prontosan^®^ wound gel	Glycerol, Hydroxyethylcellulose	Polyhexamethylene biguanide and undecylenamidopr-opyl betaine	Wound gel	Cleansing and moisturizing of skin wounds and burns	B. Braun, Melsungen, Germany
	Purilon^®^ gel, Regenecare^®^ wound gel	Collagen, aloe and sodium alginate	Lidocaine (2%)	Wound gel	Pressure ulcers, cuts, burns and abrasions	MPM Medical, Texas, USA
	Cutimed^®^ gel	Carbomer 940	Nil	Wound dresser and gel	Supports autolytic debridement in necrotic and sloughy wounds	BSN Medical, Hamburg, Germany
	Viniferamine^®^ wound hydrogel Ag	Glycerin metallic silver	Silver	Infective wound dressing	Partial and full thickness wounds with signs of infection and little to no exudate	McKesson, Texas, USA
	HemCon^®^ bandage PRO	Chitosan	Nil	Bandage	Providing hemostasis, antibacterial barrier against wide range of microorganisms	TriCol Biomedical Inc., Oregon, USA
	Hyalofill^®^-F and R	Hyaluronic acid in fleece and rope	Nil	Wound care and treatment	Absorbs wound exudate, promotes granulation tissue formulation, supports healing process	Anika, Padua, Italy
	CMC fiber dressing	Carboxymethyl cellulose	Nil	Wound dressing	Absorptive dressing for moderate to heavy exudate	Gentell, Pennsylvania, USA
	Inadine™ (PVP-1) non-adherent dressing	Polyethylene glycol	Povidone iodine	Wound dressing	Ulcers deriving from different etiologies, chronic wounds	3M Health Care Ltd., Minnesota, USA

**Table 3 pharmaceuticals-15-00371-t003:** Injectable hydrogels formulation that are under clinical trials.

Name/Sponsor Company	Gelation Mechanism	Hydrogel Material (Types)	Injection Type	Indications	Clinical Trail/Phase
Argiform (Research Centre BIOFORM, Moscow, Russia)	Chemical reaction	Polyacrylamide/silver ions (Synthetic)	Intra-articular	Knee osteoarthritis	NCT03897686 (NA)
Aquamid (Henning Bliddal, Copenhagen, Denmark)	Chemical reaction	Polyacrylamide (Synthetic)	Intra-articular	Knee osteoarthritis	NCT03060421 (NA)
PAAG-OA (Contura, Copenhagen, Denmark)	Chemical reaction	Polyacrylamide (Synthetic)	Intra-articular	Knee osteoarthritis	NCT04045431 (NA)
Aquamid (A2 Reumatologi Og Idrætsmedicin, Holte, Denmark)	Chemical reaction	Polyacrylamide (Synthetic)	Intra-articular	Knee osteoarthritis	NCT03067090 (NA)
GelStix^®^ Nucleus augmentation device (Dr med. Paolo Maino Viceprimario Anestesiologia, Germany)	Chemical reaction	Polyacrylonitrile (Synthetic)	Intra-discal	Degenerative disc disease	NCT02763956 (NA)
Hymovis Viscoelastic Hydrogel (Fidia Farmaceutici s.p.a., Italy)	Physical interaction	High molecular weight hyaluronan (Natural)	Intra-articular	Osteoarthritis	NCT01372475 (Phase III)
HYADD^®^ 4 Hydrogel (Fidia Farmaceutici s.p.a., Italy)	Physical interaction	Non-crosslinked hyaluronic acid alkylamide (Natural)	Intra-articular	Knee osteoarthritis	NCT02187549 (NA)
Promedon (Kolbermoor, Germany)	Physical interaction	Hydroxyethyl cellulose (Natural)	Knee	Osteoarthritis	NCT04061733 (NA)
Algisyl-LVR^®^ device (LoneStar Heart, Inc., California, USA)	Physical interaction	Alginate (Natural)	Intra-myocardial	Heart failure and dilated cardiomyopathy	NCT01311791 (Phase II/III)
Algisyl device (LoneStar Heart, Inc., California, USA)	Physical interaction	Alginate (Natural)	Intra-myocardial	Moderate to severe heart failure	NCT03082508 (NA)
Neo-kidney augment (inRegen, California, USA)	Chemical reaction	Gelatin with selected renal cells (Natural)	Kidney	Type 2 diabetes and chronic kidney disease	NCT02525263 (Phase II)
Renal autologous cell therapy (inRegen, California, USA)	Chemical reaction	Gelatin with renal autologous cells (Natural)	Renal cortex	Chronic kidney disease from congenital anomalies of kidney and urinary tract	NCT04115345 (Phase I)
The Second Affiliated Hospital of Chongqing Medical University (China)	Mechanism unknown	Unknown/human amniotic epithelial cells (Natural)	Uterine cavity	Asherman’s syndrome	NCT03223454 (Phase I)
Naofumi Takehara (Hiroshima, Japan)	Mechanism unknown	Gelatin with basic fibroblast growth factor (Natural)	Intra-myocardial	Ischemic cardiomyopathy	NCT00981006 (Phase I)
VentriGel (Ventrix, Inc., California, USA)	Physical interaction	Native myocardial extracellular matrix (Natural)	Trans-endocardial	Myocardial infarction	NCT02305602 (Phase I)
Absorbable Radiopaque Tissue Marker (Sidney Kimmel Comprehensive Cancer Center at Johns Hopkins, Baltimore, USA)	Chemical reaction	Polyethylene glycol/TraceIT^®^ (Synthetic)	Between pancreas and duodenum	Imaging of pancreatic adenocarcinoma	NCT03307564
Memorial Sloan Kettering Cancer Center, New York, USA	Chemical reaction	Polyethylene glycol (Synthetic)	Visceral pleura Lung	Biopsy	NCT02224924 (Phase III)
Absorbable Radiopaque Tissue Marker (Washington University School of Medicine, USA)	Chemical reaction	Polyethylene glycol/TraceIT^®^ (Synthetic)	Resection bed	Imaging of oropharyngeal cancer	NCT03713021 (Phase I)
Absorbable Radiopaque Hydrogel Spacer (Thomas, Pennsylvania, USA)	Chemical reaction	Polyethylene glycol/TraceIT^®^ (Synthetic)		Spacing in radiation therapy for rectal cancer	NCT03258541 (NA)
Augmenix, Inc. Bedford, USA	Chemical reaction	Polyethylene glycol/SpaceOAR^®^ (Synthetic)	Between the rectum and prostate	Spacing in radiation therapy for prostate cancer	NCT01538628 (Phase III)
Royal North Shore Hospital, Australia	Chemical reaction	Polyethylene glycol/SpaceOAR^®^ (Synthetic)	Between the rectum and prostate	Spacing in radiation therapy for prostate cancer	NCT02212548 (NA)
University of Washington, USA	Chemical reaction	Polyethylene glycol/TraceIT^®^ (Synthetic)	Around circumference of the tumor bed	Imaging of bladder carcinoma	NCT03125226
Gut Guarding Gel (National Cheng-Kung University Hospital, Tainan city, Taiwan)	Physical interaction	Sodium alginate/calcium lactate (Natural)	Submucosal	Gastroenterological tumor and polyps	NCT03321396 (NA)
Bulkamid (Karolinska Institutet, Stockholm, Sweden)	Chemical reaction	Polyacrylamide (Synthetic)	Transurethral	Midurethral sling surgery	NCT02776423
Bulkamid (Cantonal Hospital, Frauenfeld, Frauenfeld, Switzerland)	Chemical reaction	Polyacrylamide/botulinum toxin A (Synthetic)	Intra-vesical	Mixed urinary incontinence	NCT02815046 (NA)
Bulkamid (Contura, Copenhagen, Denmark)	Chemical reaction	Polyacrylamide (Synthetic)	Transurethral	Stress urinary incontinence	NCT00629083 (NA)
Bulkamid (Helsinki University Central Hospital, Finland)	Chemical reaction	Polyacrylamide (Synthetic)	Transurethral	Stress urinary incontinence	NCT02538991 (NA)
Bulkamid (Karolinska Institute, Huddinge, Sweden)	Chemical reaction	Polyacrylamide (Synthetic)	Submucosal	Anal incontinence	NCT02550899 (Phase IV)
Ocular Therapeutix, Inc., Massachusetts, USA	Chemical reaction	Polyethylene glycol/OTX-TKI (Synthetic)	Intra-vitreal	Neovascular age-related macular degeneration)	NCT03630315 (Phase I)
EUTROPHILL hydrogel (Assistance Publique-Hôpitaux de Paris, France)	Chemical reaction	Polyacrylamide (Synthetic)	Subcutaneous	HIV-related facial lipoatrophy	NCT01077765 (Phase III)
Frequency Therapeutics, Massachusetts, USA	Physical interaction	Poloxamer/FX-322 (Synthetic)	Intra-tympanic	Sensorineural hearing loss	NCT04120116 (Phase II)

**Table 4 pharmaceuticals-15-00371-t004:** FDA clinically approved injectable hydrogel formulations.

Brand Name/Company	Gelation Mechanism	Hydrogel Material (Types)	APIs	Injection Type	Indications	FDA Approved/Application No.
Zyplast(R)^®^ and Zyderm(R)^®^ (Inamed Corporation/Allergan, Inc., California, USA)	Chemical reaction	Bovine collagen	Bovine	Dermis	For correction of contour deficiencies	1981/FDA and EMA
Fibrel^®^ (Serono Laboratories, Geneva, Switzerland)	Physical interaction	Collagen (Natural)		Dermis	For correction of depressed cutaneous scars	1988/FDA
Fibrel^®^ (Serono Laboratories, Geneva, Switzerland)	Physical interaction	Collagen (natural)		Dermis	Correction of depressed cutaneous scars	1988/P850053
Sandostatin^®^ Novartis Pharm. Corp., Basil, Switzerland)	Temperature	PLGA	Octreotide acetate		Acromegaly	1998/021-008
Atridox^®^ Atrix Lab. Inc., London, UK	Temperature	PLGA	Doxycycline hyclate (10%)		Adult periodontitis	1998/50751
Atrisorb D^®^ Atrix Lab. Inc., London, UK	Temperature	PLGA	Doxycycline hyclate		Periodontal tissue regeneration	2000/K982865
Osteogenic protein 1(OP-1^®^) implant, OP-1^®^ Putty (Stryker Biotech, Michigan, USA)	Physical interaction	Collagen, carboxymethylcellulose, and recombinant OP-1 (Natural)		Spinal injection	Posterolateral lumbar spinal fusion	2001/FDA
INFUSE^®^ bone graft (Medtronic Sofamor Danek USA, Inc., Tennessee, USA)	Physical interaction	Collagen and recombinant human bone morphogenetic protein-2 (Natural)		Spinal injection	Spinal fusion and spine, oral-maxillofacial and orthopedic trauma surgeries	2002 for first indication/FDA
Collagen Implant, CosmoDerm^®^ 1 human-based collagen, CosmoDerm^®^ 2 human-based collagen CosmoPlast^®^ human-based collagen (Inamed Corporation/Allergan, Inc., California, USA)	Cosmo Derm: Physical interaction, Cosmo Plast: Chemical reaction	Human collagen (Natural)		Superficial papillary dermis	For correction of soft tissue contour deficiencies, such as wrinkles and acne scars	2003/FDA and EMA
Radiesse^®^ (Bioform Medical, Inc., San Mateo, USA)	Physical interaction)	Hydroxylapatite, carboxymethyl-cellulose (Synthetic)		Dermis	For correction of facial folds and wrinkles, signs of facial fat loss and volume loss	2004/EMA 2006/FDA (for first indication)
UFLEXXA^®^ (Ferring Pharmaceuticals Inc., Saint-Prex, Switzerland)	Physical interaction	Hyaluronic acid (Natural)		Intra-articular	Knee osteoarthritis)	2004/FDA 2005/EMA
Hylaform^®^ (Hylan B gel), Captique Injectable Gel, Prevelle Silk (Genzyme Biosurgery., Massachusetts, USA)	Chemical reaction)	Modified hyaluronic acid derived from a bird (avian) source (Natural)		Dermis	Correction of moderate to severe facial wrinkles and folds	1995/EMA 2004/FDA
Sculptra^®^ (Sanofi Aventis, New Jersy, USA)	Physical interaction	Poly-L-lactic acid (Synthetic)		Dermis	For correction of signs of facial fat loss, shallow to deep contour deficiencies and facial wrinkles	2000/EMA 2004/FDA (for first indication)
Coaptite^®^ (BioForm Medical, Inc., San Mateo, USA)	Physical interaction	Calcium hydroxylapatite, sodium carboxymethylcellulose, glycerin (Synthetic)		SC	Female stress urinary incontinence)	2001/EMA 2005/FDA
Artefill^®^ (Suneva Medical, Inc., California, USA)	Physical interaction	Polymethylmethacrylate beads, collagen and lidocaine (Synthetic)		Dermis	Facial wrinkles and folds	2006/FDA
Juvéderm^®^/Voluma XC/Ultra XC/Volbella XC/Vollure XC (Allergan, Inc., California, USA)	Chemical reaction	Hyaluronic acid (Natural)		Facial tissue, cheek, lips	For correction of facial wrinkles and folds, volume loss and lip augmentation. EMA (2000) FDA (2006 for first indication)	2000/EMA 2006/FDA (for first indication)
Bulkamid^®^ hydrogel (Searchlight Medical Inc., New York, USA)	Chemical reaction	Polyacrylamide		Transurethral	Female stress urinary incontinence	2003/EMA 2006/FDA
Elevess^®^ (Anika Therapeutics, Massachusetts, USA)	Chemical reaction	Hyaluronic acid with lidocaine (Natural)		Dermis	Moderate to severe facial wrinkles and folds	2006/FDA 2007/EMA
Supprelin LA^®^ (Indevus Pharmaceuticals, Inc., Massachusetts, USA)	Chemical reaction	Histrelin acetate, Poly (2-hydroxyethyl methacrylate) (Synthetic)		SC	Central precocious puberty	2005/EMA 2005/FDA
Evolence^®^ Collagen Filler (Colbar Lifescience, Herzliya, Israel)	Chemical reaction	Collagen (Natural)		Dermis	Moderate to deep facial wrinkles and folds	2004/EMA 2008/FDA
Belotero Balance^®^ (Merz Pharmaceuticals., Frankfurt, Germany)	Chemical reaction	Hyaluronic acid (Natural)		Dermis	Moderate to severe facial wrinkles and folds	2004/EMA 2011/FDA
Juvéderm^®^ XC (Allergan, Inc., California, USA)	Chemical reaction	Hyaluronic acid with lidocaine (Natural)		Facial tissue	Correction of facial wrinkles and folds	2010/FDA
SpaceOAR^®^ Hydrogel (Augmenix, Inc., Massachusetts, USA)	Chemical reaction	Polyethylene glycol (Synthetic)		Percutaneous	For protecting vulnerable tissues during prostate cancer radiotherapy	2010/EMA 2015/FDA
Restylane^®^ Lyft, Restylane^®^ Refyne, Restylane^®^ Defyne (Galderma Laboratories, L.P., Texas, USA) Restylane^®^ Silk (Valeant Pharmaceuticals North America LLC/Medicis, USA) Restylane^®^ Injectable Gel (Medicis Aesthetics Holdings, Inc., New Jersy, USA)	Chemical reaction	Hyaluronic acid with Lidocaine (Natural)		SC, dermis, lips	For correction of volume deficit, facial folds and wrinkles, midface contour deficiencies and perioral rhytids	2010/EMA 2012/FDA (for first indication)
TraceIT^®^ Hydrogel Tissue Marker (Augmenix, Inc., Massachusetts, USA)	Chemical reaction	Polyethylene glycol (Synthetic)		Percutaneous	Improved soft tissue alignment for image guided therapy	2013/FDA
Algisyl-LVR^®^ Hydrogel Implant (LoneStar Heart, Inc., California, USA)	Physical interaction	Alginate (Natural)		Percutaneous	Advanced heart failure	2014/EMA
Vantas^®^ (Endo Pharmaceuticals., Pennsylvania, USA)	Chemical reaction	Histrelin acetate, poly (2-hydroxyethyl methacrylate), poly(2-hydroxypropyl methacrylate) and gonadotropin releasing hormone (Synthetic)		SC	Palliative treatment of prostate cancer	2004/FDA2005/EMA
Radiesse^®^ (+) (Merz Pharmaceuticals., Frankfurt, Germany)	Physical interaction	Hydroxylapatite, carboxymethyl-cellulose with Lidocaine (Synthetic)		Dermis	Correction of wrinkles and folds, stimulation of natural collagen production	2015/FDA
Teosyal^®^ RHA (Teoxane SA., Geneva, Switzerland)	Chemical reaction	Hyaluronic acid (Natural)		Dermis	Facial wrinkles and folds	2015/EMA 2017/FDA
Revanesse^®^ Versa/Revanesse^®^ Ultra (Prollenium Medical Technologies Inc., Aurora, Canada)	Chemical reaction	Hyaluronic acid (Natural)		Dermis	Moderate to severe facial wrinkles and creases	2017/FDA
Revanesse^®^ Versa., California, USA	Chemical reaction	Hyaluronic acid with lidocaine (Natural)		Dermis	Moderate to severe facial wrinkles and creases	2018/FDA
Belotero balance^®^ (+) Lidocaine (Merz Pharmaceutical., Frankfurt, Germany)	Chemical reaction	Hyaluronic acid with lidocaine (Natural)		Dermis	Moderate to severe facial wrinkles and folds	2019/FDA

**Table 5 pharmaceuticals-15-00371-t005:** Injectable hydrogels synthesized through chemical and physical crosslinking methods, their application and limitations.

Chemical/Physical Crosslinking	Types of Hydrogel Material	Hydrogel Synthesis Procedure	Applications and Advantages	Limitations and Disadvantages	Reference
Hydrophobic interaction	Hydrophilic monomers and hydrophobic co-monomers	Free radical copolymerization of a hydrophilic monomer with a hydrophobic co-monomer	Absence of crosslinking agents and relative ease of production	Poor mechanical characteristics	[65]
Ionic interaction	Solution and multivalent ions of opposite charge	Polyelectrolyte ionic interaction through simple ion exchange mechanisms and complex formation	Crosslinking takes place at room temperature and physiological pH Properties can be fine-tuned by cationic and anionic constituents	Limited to ionic polymers and sensitive to impurities	[66]
Hydrogen bond	Polymeric functional groups of high electron density with electron-deficient hydrogen atom	Self-assembly through secondary molecular interactions	Increase in polymer concentration can increase the stability of gel	Influx of water can disperse/dissolve the gel within short duration	[67]
Bulk polymerization	Monomers and monomer-soluble initiators	The polymerization reaction is initiated with radiation, ultraviolet or chemical catalysts at low rate of conversions	A simple and versatile technique for preparing hydrogels with desired physical properties and forms	Increase in viscosity during high rate of polymerization reaction can generate heat Weak polymer structure	[68]
Solution polymerization	Ionic or neutral monomers with the multifunctional crosslinking agent	Reaction initiated thermally with UV irradiation or by redox initiator system	Control of temperature Performed in non-toxic aqueous medium at room temperature High polymerization rate	To be washed to eliminate reactants, the polymers and other impurities	[69]
Suspension polymerization	Hydrophilic monomers, initiators, cross-linkers and suspending agent	The monomers and initiator are dispersed in the organic phase as a homogenous mixture	Directly usable as powders, beads or microspheres Restricted to water insoluble polymer	Cooling jacket required to dissipate heat Requirement of agitators and dispersant	[70]
Grafting	Viny polymers, initiators and crosslinking agents	Covalent bonding of monomers on free radicals generated on stronger support structures	Improve functional properties of the polymer	Difficulty of characterizing side chains	[71]
Irradiation	High energy gamma beams and electron beams as initiators	Irradiation of aqueous polymer solution results in the formation of radicals and macroradicals on the polymer chains	Pure, sterile, residue-free hydrogel Does not require catalyst and other additives Irradiation dose can control swelling capacity	Irradiation can cause polymer degradation via chain scission and crosslinking events	[72]
Step growth polymerization	Bi- or multifunctional monomers and each with attest two sites for bonding	Multifunctional monomers react to form oligomers resulting in long chain polymers	No initiator is required to start the polymerization and termination reactions	Prolonged reaction times required to achieve a high degree of conversion and high molecular weights	[73]

## Data Availability

Data sharing not applicable.

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
