# Peer review of "Current and Future Prospective of Injectable Hydrogels—Design Challenges and Limitations"

_pharmaceuticals, 2022, doi:10.3390/ph15030371_

Round 1

Reviewer 1 Report

In their manuscript, authors review the current and future prospective of injectable hydrogels with associated challenges and limitations. Authors have reviewed in detail their classification crosslinking strategies, physical properties and state of art at present.  They briefly summarize the various commercially available hydrogel and their characteristics. They then discuss the current trends and the applications of IHs in drug delivery in protein, vaccine, DNA-gene delivery, tissue engineering, and regenerative medicine. Authors are reviewed the therapeutic application of hydrogels in various field like can therapy, wound healing and bone regeneration. I really like that authors have compiled an extensive list of IHs under clinical trials, which is noteworthy. In the next sections, authors tabulated a comprehensive list of chemical and physical crosslinking methods and what are the shortcoming in each method, which also give reader a better perspective. Lastly, they described in detail the various challenges while formulating injectable hydrogels, reaging form mechanical properties till the regulatory approvals. I thoroughly enjoyed reading the review article and really like the manuscript and it is very well written. I would like to congratulate the authors to compile massive studies in one document. Overall, the review a very interesting and have one minor comment which should be addressed:

Minor comment:

Legend in Figure 7 is mislabeled please make corrections

Author Response

Reviewer 1 Comments: Legend in Figure 7 is mislabeled

Author Response to Reviewer 1

The authors are thankful to our valuable reviewer for his valuable suggestions and appreciation. The respectable reviewer has raised and did notice our typo mistakes, in the legend of Figure 7, we did notice and correct properly in our main manuscript, track changes can be followed for proofreading.

Reviewer 2 Report

The review deals with the injectable hydrogels with their past, today and future perspectives. The authors included the classification, synthesis, and characterization of injectable hydrogels, moreover, their commercial products on the market. The review is well prepared, supported by figures and lists of the hydrogels. It can be published after some minor corrections as follows :

C1) Please add the related references under the figures which were taken from other articles. The attribution such as “Copyright © 2015 European Polymer Journal with modification” would not be enough and referring to not a certain publication.  

C2) Figure 5 seems like a table. I suggest presesent it as a table. 

C3) Line 660 : “2.2.4.. Tissue-Engineering” has double dots. Please remove one of those. 

C4) Line 931: “3.2.6. Regulatory approvals”.  It is supposed to be 3.2.7. Please correct it.

Author Response

C1) Please add the related references under the figures which were taken from other articles. The attribution such as Copyright © 2021 Polymer MDPI with minor modification would not be enough and referring to not a certain publication.

Response to Reviewer 2: The authors are thankful to our respectable reviewer for his valuable comments. Yes, we did address, cite each Figure which was taken from their respective journal, for proof, track changes can be followed.

C2) Figure 5 seems like a table. I suggest present as a table. 

Response to Reviewer 2: Yes, we did honor the valuable suggestions and revise Figure 5, and changed it into Table 1, for more reference follow our track changes.

C3) Line 660: “2.2.4.. Tissue engineering” has double dots. Please remove one of those.

Response to Reviewer 2: Yes, we did notice this typo mistake, and double dots were removed as per valuable suggestions from our respectable reviewer.

C4) Line 931: “3.2.6. Regulatory approvals”. It is supposed to be 3.2.7. Please correct.

Response to Reviewer 2: Yes, it was also a typo mistake, we did notice and correct it per valuable suggestions from our respectable reviewer. Track changes for confirmation.